



# Technical Note: A double-Manning approach to compute robust rating curves and hydraulic geometries

Andrew D. Wickert[1,2,3], Jabari C. Jones[1,2,4], and Gene-Hua Crystal Ng[1,2]

[1]Saint Anthony Falls Laboratory, University of Minnesota, Minneapolis, MN 55414, USA
[2]Department of Earth and Environmental Sciences, University of Minnesota, Minneapolis, MN 55455, USA
[3]Sektion 4.6: Geomorphologie, Deutsches GeoForschungsZentrum (GFZ), Potsdam, 14473, Germany
[4]Department of Earth and Oceanographic Science, Bowdoin College, Brunswick, ME 04011, USA

**Correspondence:** Andrew Wickert (awickert@umn.edu)

**Abstract.**

Rating curves describe river discharge as a function of water-surface elevation ("stage"). They are applied globally for stream monitoring, flood-hazard prediction, and water-resources assessment. Because most rating curves are empirical, they typically require years of data collection and are easily affected by changes in channel hydraulic geometry. Here we present a

straightforward strategy based on Manning's classic equation to address both of these issues. This "double-Manning" approach employs Manning's equation for flow in and above the channel. Flow across the floodplain can follow either a Manning-inspired power-law relationship or, in the common case of a rectangular floodplain and valley-wall geometry, a second application of Manning's equation analogous to that applied within the channel. When applied to ample data from established stream gauges, we can solve for Manning's $n$ for in-channel flow, channel-bank height, and two floodplain-flow variables. When applied to

limited discharge data from a field campaign, additional constraints from the surveyed floodplain cross section permit a fit to the double-Manning formulation that matches ground truth. Using these double-Manning fits, we can dynamically adjust the rating curve to account for evolution in channel width, depth, and/or slope, as well as in channel and floodplain roughness. Such rating-curve flexibility, combined with a formulation based in flow mechanics, enables predictions during times of coupled hydrologic–geomorphic change. Open-source software with example implementations is available via GitHub, Zenodo, and

PyPI.

## 1 Introduction

Hydrologists routinely measure river stage (water-surface elevation above an arbitrary datum) and convert it into discharge (the volume of water passing through a cross section per unit time) (World Meteorological Organization, 2010b). Both variables are useful. Stage informs flood hazard and can be used together with streambed elevation to compute water-induced shear stresses

that produce turbulence, transport sediment, and shape the stream – including its aquatic habitat (Van Steeter and Pitlick, 1998; Pitlick and Van Steeter, 1998; Luppi et al., 2009; Buahin et al., 2017). Discharge informs water supply and can be linked with catchment-scale hydrological water balances and processes (e.g., Dobriyal et al., 2017; Gore and Banning, 2017; Hut et al., 2022).





Measuring stage is generally much easier than measuring discharge, though discharge is often of more hydrological interest
(World Meteorological Organization, 2010a). Therefore, hydrologists have developed "rating curves" – mathematical expressions for discharge as a function of stage – for rivers around the world. Most such stage–discharge relationships are fit using a power-law function with a stage offset (e.g., Leopold and Maddock, 1953; Petersen-Øverleir, 2005; Schmidt and Yen, 2008; World Meteorological Organization, 2010b; Hamilton et al., 2016; Hrafnkelsson et al., 2022):

$$Q = k \left( z_s - z_b \right)^P. \tag{1}$$

Here, $Q$ is water discharge, $k$ is an empirically derived coefficient, $z_s$ is stage, $z_b$ is the stage at which $Q = 0$ (measured from the same datum as $z_s$; subscript $b$ indicates the channel bed), and $P$ is an empirically derived exponent.

The power-law form of Equation 1 generalizes Manning's equation, a longstanding solution for turbulent open-channel flow velocity (Manning, 1891; Gioia and Bombardelli, 2001; Bonetti et al., 2017), to include the ability to adjust (via its coefficient and exponent) to a range of hydraulic geometries (e.g., Petersen-Øverleir, 2005; Schmidt and Yen, 2008; World Meteorological
Organization, 2010b). This power-law method may be extended through segmentation into or summation of multiple power-law functions (Petersen-Øverleir and Reitan, 2005; Le Coz et al., 2014; Hodson et al., 2024). Best-fitting curves and their associated error may be solved with the support of Bayesian inference (Moyeed and Clarke, 2005; Petersen-Øverleir and Reitan, 2005; Le Coz et al., 2014; McMillan et al., 2017; Kiang et al., 2018; Hodson et al., 2024), which improves uncertainty quantification. For example, the BaRatin (Bayesian Rating Curves) toolset (Le Coz et al., 2014, 2024) can formulate arbitrarily
complex best-fitting rating curves.

Here we present a "double-Manning" formulation as a specific power-law-segmented rating curve comprising (1) Manning's equation for flows within and above a rectangular channel and (2) either a Manning-inspired power law or a second instance of Manning's equation for overbank flows (i.e., those crossing floodplains). By adding basic flow mechanics (Gioia and Bombardelli, 2001; Bonetti et al., 2017), this double-Manning approach follows a "middle road" between fully empirical
(and semi-empirical) power-law approaches and more complex hydrodynamic models (e.g., Pizzuto, 1991; Brunner, 1995; Kean and Smith, 2005, 2010; Quintero et al., 2021). Furthermore, the double-Manning approach transforms free parameters – variables that require calibration – into explicitly defined and field-measurable channel and floodplain properties. This adds information to traditionally used stage–discharge data when constructing rating curves. It also permits direct simulation of changing channel conveyance capacity response to changes in channel form (e.g., Slater et al., 2015; Ahrendt et al., 2022).
We implement the double-Manning formulation into the `doublemanning` software package (Wickert, 2023). This open-source module follows the conventions of the Community Surface Dynamics Modeling System (CSDMS) (Peckham et al., 2013; Overeem et al., 2013; Tucker et al., 2022) to facilitate direct coupling with relevant models (e.g., of hydrologic and geomorphic processes). In addition, the double-Manning formulation can be emulated using BaRatin (Le Coz et al., 2014, 2024), which may be useful if a Bayesian approach is desired. The double-Manning software is available from GitHub, Zenodo, and
PyPI (Wickert, 2023).



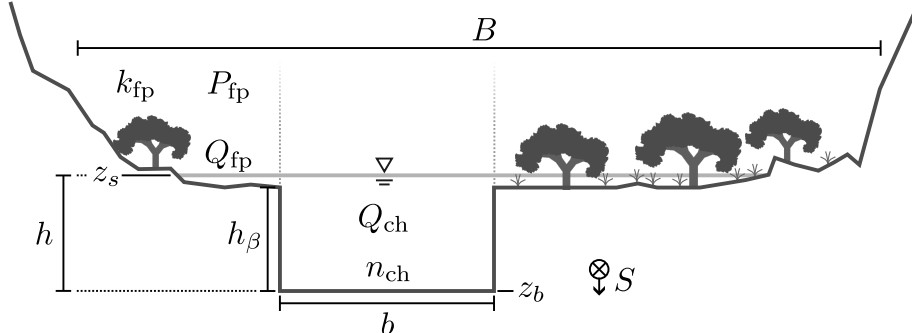

**Figure 1.** Stream cross-section schematic of the double-Manning approach to relating river stage and discharge. A rectangular channel with width $b$ carries the flow until it goes overbank (above the channel-bank height $h_\beta$), at which point it enters the floodplain on either side, whose topography and roughness structure may be more arbitrarily complex. Channel discharge, $Q_{ch}$, encompasses flow within and (if the water level is high enough) above the channel, as suggested by the fading dotted lines. Floodplain discharge, $Q_{fp}$, comprises flow to both the left and right of the channel. Other variables: $B$: valley-bottom width. $P_{fp}$: floodplain-flow exponent. $k_{fp}$: floodplain-flow coefficient. $z_s$: river stage. $h$: water depth above the channel bed. $n_{ch}$: Manning's $n$ for flow resistance within the channel. $z_b$: channel-bed stage – that is, the flow stage at which discharge is 0. $S$: channel slope (with the symbol indicating downslope). Full discharge $Q = Q_{ch} + Q_{fp}$.

## 2 Double-Manning Formulation

### 2.1 Channel

For the channel and the region directly above it (Figure 1), we use the classic Manning (1891) formula to solve for velocity as a function of flow depth,

$$\bar{u} = \frac{1}{n_{ch}} R_h^{2/3} S^{1/2}. \tag{2}$$

Here, $\bar{u}$ is depth-averaged velocity, $n_{ch}$ is Manning's roughness coefficient within the channel, and $S$ is the downstream-directed channel slope. $R_h$ is the hydraulic radius, which is defined as the ratio of the cross-sectional area to the wetted perimeter.

Although channels may have complex geometries, a rectangular channel approximation appears reasonable across a range of fluvial geomorphic studies (e.g., Parker, 1979; Naito and Parker, 2019; Wickert and Schildgen, 2019); therefore,

$$R_h = \frac{bh}{b + 2(h \wedge h_\beta)}, \tag{3}$$

where $b$ is channel width, $h$ is flow (i.e., water) depth, $h_\beta$ is the height of the channel banks, and $\wedge$ indicates that the smaller of the two numbers be taken. Therefore, the wetted perimeter is computed using the flow depth when the water level lies below the height of the banks, and by the bank height when it is at or above these. The rectangular channel approximation simplifies data needs because it can be applied using straightforward field or remotely sensed observations of channel width and bank height.





This wetted-perimeter definition neglects drag from the floodplain that impacts flow velocities above the channel. This assumption is reasonable so long as the channel is much wider than the overbank flows are deep. Most natural channels and floods satisfy this criterion (see Dunne and Jerolmack, 2020, for a discussion of channel aspect ratios).

Flow depth, $h$, relates to stage, $z_s$, as:

$$h = z_s - z_b. \tag{4}$$

This follows the convention that stage is height above an arbitrary datum, which does not generally equal the river-bed elevation, $z_b$.

Following this rectangular channel approximation, we multiply in-channel velocity by channel cross-sectional area to com-
pute discharge:

$$Q_{\text{ch}} = \bar{u}bh. \tag{5}$$

Expanding $\bar{u}$,

$$Q_{\text{ch}} = \frac{b}{n_{\text{ch}}} h R_h^{2/3} S^{1/2}. \tag{6}$$

### 2.1.1 Wide-channel approximation

Optionally, one may use the double-Manning approach with the wide-channel approximation – that is, with $b \gg h_\beta$ and there-
fore $h \approx R_h$ across all flow depths. Under this simplification, Equation 6 becomes

$$Q_{\text{ch}} = \frac{b}{n_{\text{ch}}} h^{5/3} S^{1/2}. \tag{7}$$

The wide-channel approximation is available as an option in the `doublemanning` software package (Wickert, 2023). For the remainder of this paper, we use Equation 6 (rather than Equation 7) because it more accurately represents the physical system
and is appropriate for a wider range of river geometries.

### 2.2 Floodplain

Flow across floodplains requires a separate but related mathematical expression from that for flow within and above the channel. Once water enters the floodplain, it interacts with different roughness elements. Furthermore, if the floodplain is not horizontal and planar, increasing water depths will correspond to increasing flow widths.

In order to represent the complexity of floodplain flow in a simple mathematical form, we generalize Equation 2 (Manning, 1891) into a generic power law, as is commonly done for stage–discharge rating curves (Equation 1). This power-law form includes a coefficient, $k_{\text{fp}}$, that increases with floodplain width and decreases with floodplain roughness due, e.g., to vegetation. It also includes an exponent, $P_{\text{fp}}$, that relates to floodplain topography and systematic spatial variability within the roughness structure:

$$Q_{\text{fp}} = k_{\text{fp}} \left( h - h_\beta \right)^{P_{\text{fp}}}. \tag{8}$$





Here we use overbank flow depth, $h - h_\beta$, because $k_{\mathrm{fp}}$ and $P_{\mathrm{fp}}$ can include the effects of valley-wall drag, thereby implicitly incorporating floodplain hydraulic radius. Furthermore, we posit that the inundation width and depth distributions can be described with power-law functions.

### 2.2.1 Explicit Manning's equation for the floodplain

For the simplified case in which the floodplain may be approximated to have a rectangular form that is much wider than it is deep, one may apply the wide-channel approximation (see Equation 7) to the floodplain and therefore rewrite Equation 8 as

$$Q_{\mathrm{fp}} = \frac{(B-b)}{n_{\mathrm{fp}}} (h - h_\beta)^{5/3} S^{1/2}. \tag{9}$$

Here, $B$ is the width of the valley bottom – that is, the combined floodplain and channel. $n_{\mathrm{fp}}$ is the Manning (1891) roughness coefficient for the floodplain surface. This use of Manning's equation for the floodplain in addition to the channel prompted

our use of "double-Manning" to refer to our approach, even though our more general formulation (Equation 8) implements a power law whose exponent is a variable. Linking Equation 9 to Equation 8,

$$k_{\mathrm{fp}} = \frac{(B-b)S^{1/2}}{n_{\mathrm{fp}}} \tag{10}$$

and

$$P_{\mathrm{fp}} = \frac{5}{3}. \tag{11}$$

The wide-rectangular floodplain approximation applied here may not always be appropriate: some rivers have floodplains that are only a few channel widths wide. However, many floodplains contain such significant internal roughness (e.g., from vegetation) that the additional drag against their side walls is small in comparison. Therefore, we consider the wide-rectangular floodplain approximation to be reasonable even when not formally defensible based on Equation 3 alone.

### 2.3 Full equation

Overall river discharge is the sum of flows within and above the channel, $Q_{\mathrm{ch}}$, and those (if any) that cross the floodplain, $Q_{\mathrm{fp}}$ (Figure 1):

$$Q = \begin{cases} Q_{\mathrm{ch}} & \text{if } h \leq h_\beta \\ Q_{\mathrm{ch}} + Q_{\mathrm{fp}} & \text{otherwise.} \end{cases} \tag{12}$$

$Q_{\mathrm{ch}}$ is defined in Equation 6. For wide and approximately rectangular floodplains, Equation 9 can be used for $Q_{\mathrm{fp}}$. More generally, Equation 8 can be applied for $Q_{\mathrm{fp}}$.

## 3 Data constraints


We formulated the double-Manning approach to have parameters whose values are straightforward and/or easy to measure. Including these parameters facilitates rating-curve development where discharge measurements may be limited. Alternatively,



**Table 1.** Parameters for the double-Manning approach and their connection to observations that are independent of stage–discharge data. Observation (Obs.) difficulty qualitatively integrates the difficulty of the measurement, the logistics involved in making it, and how directly the measurement yields a quality estimate for the desired parameter.

| Parameter | Variable | Observation method(s) | Obs. difficulty |
|---|---|---|---|
| Channel width [m] | $b$ | Overhead photos, DEMs[a], and/or field surveys | Very Easy |
| Channel slope | $S$ | DEMs[a], possibly aided by overhead photos, or field surveys | Easy |
| Channel depth (bank height) [m] | $h_\beta$ | Gauge-survey data, DEMs, field surveys | Easy |
| Stage Offset: Stage at Q = 0 [m] | $z_b$ | Approximate mean bed elevation from field surveys | Easy |
| In-channel Manning's $n$ | $n_{\text{ch}}$ | Grain size; Manning's $n$ tables or photos | Intermediate |
| Floodplain Manning's $n$ | $n_{\text{fp}}$ | Manning's $n$ tables or photos | Intermediate |
| Floodplain discharge coefficient | $k_{\text{fp}}$ | Insight from field surveys or Manning's $n$ tables or photos | Hard |
| Floodplain discharge exponent | $P_{\text{fp}}$ | Insight from topographic and roughness surveys of floodplain | Hard |

[a] Digital elevation models

when there are sufficient discharge-stage observations to estimate these parameters through inversion, the results can be checked against measurements (Section 4.2).

## 3.1 Field observations

Table 1 holds the parameters alongside information on observation methods and difficulty. This "difficulty" reflects three factors. The first is the ease (including the time required) to make the measurement itself. Stretching a measuring tape across a streambank would be easy, wading in a river to measure velocity would be a bit more difficult, and (due to the time required) surveying all floodplain vegetation for stem density (to parameterize roughness) would be much harder. The second factor relates to logistics: it is easier to make an observation from remotely sensed data or archived gauging station notes than it is to go to the field. The third factor indicates how directly the variable in question may be measured. Geometric and velocity data are measured directly, whereas estimates of in-channel Manning's $n$ require calculations or inferences from gravel grain size, bedforms, and/or woody debris. Floodplain parameters may be even harder to directly estimate, as they relate to flow paths and vegetation distributions. These, therefore, may be easier to back-calculate from stage–discharge data (e.g., Section 5.1).

Measurements of channel width and bank height (i.e., channel depth) link to discharge through the rectangular channel approximation. Therefore, the approximate "rectangular-channel" values for both width and depth should be selected with flow mechanics in mind (e.g., Naito and Parker, 2019). For shallow but wide flows, $Q \propto h^{5/3}$. Because of this exponent greater than 1, the discharge difference between a shallow flow and no flow can be much less than that between the same shallow flow and a flow of twice its depth. Therefore, the effective channel depth for the double-Manning approach will be less than the distance from the floodplain surface to the thalweg. Channel width should be estimated above low-lying bars, as their inundation will occur with only a small change in discharge.





In addition to considering the ease of field measurement, the double-Manning parameters (Table 1), appearing in Equations 6 and 8, indicate which observations most significantly constrain the double-Manning solution. Equations 9 and 10 provide additional context when the wide-rectangular floodplain approximation (Section 2.2.1) may be reasonably applied. We discuss 150 these considerations briefly in Sections 3.2 and 3.3.

### 3.2 Constraining in-channel parameters

In Equation 6, discharge depends on channel width ($b$) and in-channel Manning's $n$ ($n_{\text{ch}}$). However, because it depends primarily on their ratio, estimating these two parameters independently from each other using just stage–discharge measurements would be difficult. (Although $R_h$ also includes dependence on $b$, most channels are wide relative to their depth, making $R_h$ 155 much more sensitive to $h$ than to $b$.) Fortunately, $b$ is the easiest variable to measure, and this often can be done without field work (Table 1). Some care must be taken to ensure that the width of the approximately rectangular channel is made, rather than the width of the flow at the time of the observation.

Similarly straightforward measurements can provide slope ($S$) for Equation 6. This can be done with digital elevation models, which are available for most of the world, and/or via field surveys. Slope should be measured over a long enough 160 distance to provide a reach-scale mean that averages across local variability in stream topography.

Field observations of grain size, bed configurations, and surface roughness may be used to estimate $n_{\text{ch}}$. In sand-bed rivers, $n_{\text{ch}}$ commonly varies within the range $\sim$0.018–0.040 (Simons et al., 2004). Hey (1979), followed by a further analysis by Clifford et al. (1992), demonstrated that for gravel-bed rivers, the Nikuradse (1933) roughness parameter $k_s$ can be determined by $k_s \approx 3.5 D_{84}$, where $D_{84}$ is the 84th percentile of the grain-size distribution. Combining the work of Parker (1991) with the 165 definition of Manning's $n$ provides an algebraic translation between $k_s$ and $n_{\text{ch}}$:

$$n_{\text{ch}} = \frac{k_s^{1/6}}{g^{1/2}\alpha_r}. \tag{13}$$

Here, $g$ is acceleration due to gravity and $\alpha_r$ is a coefficient that increases as the flow becomes faster for a given $k_s$. For gravel-bed rivers, Parker (1991) found that $\alpha_r \approx 8.1$. Entering these constants and the above approximation for $k_s$ yields

$$n_{\text{ch}} \approx 0.049 \ \text{s}\,\text{m}^{-1/2} \times D_{84}^{1/6}, \tag{14}$$

where s and m are the units "second" and "meter". Fundamentally, Equation 14 could also provide $n_{\text{fp}}$, but floodplain vegetation and topography can often provide more significant roughness than that provided from gravel grain size.

### 3.3 Constraining floodplain parameters

Estimates of channel-bank height ($h_\beta$) provide an expected flow stage at which the rating curve should "roll over". As flow rises above this stage, it enters the floodplain: at this point, a modest increase in stage may require a large increase in discharge. The 175 double-Manning formulation explicitly incorporates this rating-curve feature via its piecewise form in Equation 12. Although channel-bank height may be solved for as a free parameter using a large amount of stage–discharge data (Section 5.1), situations involving sparse (or no) data are aided by (or require) independent estimates of bank height (Sections 5.2 and 5.3).





If floodplain geometry and roughness variability are unknown, then $P_{\text{fp}}$ and $k_{\text{fp}}$ must be left to calibration based on stage–discharge measurements. To aid parameter estimation, we turn to features of floodplain topography. Floodplains have generally

concave-up forms, from flat regions near the river that rise to terraces or valley walls farther from the channel. Such floodplain forms produce greater flow width – and therefore discharge – as stage increases. Although river levees and abandoned bars do comprise local convexities (e.g., Moody et al., 1999; Hassenruck-Gudipati et al., 2022), these in fact amplify the overall trend towards wider flow occupation as water rises. Therefore, $P_{\text{fp}} = 5/3$, corresponding to a wide rectangular floodplain cross-sectional geometry, provides a reasonable lower bound for the floodplain-flow exponent.

The special case of the wide rectangular floodplain constrains $P_{\text{fp}} = 5/3$. Solutions using the wide-rectangular floodplain approximation must also incorporate valley-floor width, $B$, which may be measured remotely or through field surveys (Table 1). With $P_{\text{fp}}$ fixed and $B$ known, $n_{\text{fp}}$ (which may be estimated in the field or based on literature values) may be extracted from the bulk coefficient, $k_{\text{fp}}$ (Equation 10).

## 4    Numerical implementation and inversion

### 4.1    Model implementation and interface

We implement our double-Manning formulation into the `doublemanning` software package. This contains two key modules. The first, `doublemanning-fit`, computes an optimized parameter set (Table 1) to fit Equation 12 (including Equation 6 and either Equation 8 or Equation 9) to stage–discharge data. `doublemanning-calc` computes flow depth or stage as a function of discharge, as well as discharge as a function of flow depth or stage. The `doublemanning` package is available via

GitHub, archived on Zenodo (Wickert, 2023), and may be downloaded using `pip` from PyPI. This `pip`-based install includes command-line interfaces for both `doublemanning-fit` and `doublemanning-calc`.

Users may access both `doublemanning-fit` and `doublemanning-calc` using self-documented command-line interfaces. For `doublemanning-fit`, channel width ($b$), depth ($h_{\beta}$), and/or slope ($S$) may be passed directly via the command-line interface. To additionally set optimization upper and lower bounds on $n_{\text{ch}}$, $k_{\text{fp}}$, $P_{\text{fp}}$, $z_b$, $h_{\beta}$, and/or $b$, a user may format a

YAML file following the guide within the double-Manning repository (Wickert, 2023). We recommend working with YAML files, as these give additional control over plotting and allow the inputs and constraints on the fits to be self-documented.

In addition to this command-line interface, users may interact with `doublemanning` as a Python module. Its function structure follows CSDMS standards (Peckham et al., 2013; Overeem et al., 2013; Tucker et al., 2022). One may use it to produce and update rating curves and to translate between stage and discharge as a standalone module or as part of a coupled

hydraulic–geomorphic modeling system.

### 4.2    Inverse modeling: fitting stage–discharge data

We designed the double-Manning approach to permit multiple links with field data while limiting the required parameters to a modest set, most of which can be readily measured (see Ben-Zion, 2017, for a discussion of model explanatory power vs.





complexity). Equation 12 involves four field-measurable parameters: channel width($b$), bank height ($h_\beta$), channel-bed elevation
($z_b$), and in-channel Manning's $n$ ($n_{\mathrm{ch}}$). It also includes two free parameters requiring selection or calibration, the power-law
coefficient ($k_{\mathrm{fp}}$) and exponent ($P_{\mathrm{fp}}$) for flows across the floodplain, which relate to floodplain topography and roughness.

The `doublemanning-fit` software module finds an optimized parameter set (see Table 1) to fit Equation 12 to user-
provided stage–discharge data. It does so via a nonlinear least-squares approach using the `curve_fit` method within SciPy
(Virtanen et al., 2020). Users can specify values for width ($b$), depth ($h_\beta$), and/or slope ($S$); they may also specify bounds for
in-channel Manning's $n$ ($n_{\mathrm{ch}}$), the floodplain coefficient ($k_{\mathrm{fp}}$) and/or exponent ($P_{\mathrm{fp}}$), and the offset between flow depth and
river stage ($z_b$). By directly setting (or setting tight estimation bounds for) these parameters, one can impose known constraints
on the system, including the wide-rectangular floodplain approximation (Equation 9). By loosening these, one can find values
that are harder to determine independently (e.g., Manning's $n$ or, harder still, the floodplain parameters). One may also loosen
the bounds on observed parameters to test that the `doublemanning-fit` method recovers expected values.

## 4.3 Forward modeling

The `doublemanning-calc` module uses the output best-fitting parameters from `doublemanning-fit` to convert be-
tween water depth ($h$) and discharge ($Q$). Directly solving Equation 12 provides discharge as a function of water depth. This
same solution method provides discharge as a function of stage ($z_s$), following a first-step subtraction of $z_b$ (Equation 4). Us-
ing Equation 12 to calculate water depth ($h$) from discharge ($Q$) is not as trivial because $h$ cannot be isolated on one side of
the equation. Therefore, we obtain $h$ and (through trivial addition) $z_s$ from $Q$ using the `fsolve` root-finder built into SciPy
(Virtanen et al., 2020).

`fsolve` is a Python frontend to the FORTRAN MINPACK library (Moré et al., 1980), which implements Powell's (1970)
dog-leg method for iteratively solving non-linear least-squares problems. Powell's dog-leg method works by applying the
Gauß–Newton algorithm so long as the solution lies within an imposed "trust region" of permissible parameter values, and
otherwise augments this approach with a forward-difference-style gradient descent.

## 5 Field Applications

We apply this double-Manning approach to three rivers (Table 2) to demonstrate its applicability across a wide range of settings
and quantities of available data. From largest to smallest, these are the sand-bedded Minnesota River, gravel-bedded Cannon
River, and gravel-to-boulder-bedded La Dormida. These rivers span nearly three orders of magnitude in discharge, four orders
of magnitude in slope, and three orders of magnitude in bed-material grain size. Each river has a floodplain.

The Minnesota and Cannon Rivers (Minnesota, USA) host long-term United States Geological Survey (USGS) stream
gauges. We analyze a data set from the Minnesota River spanning 1934–2021, providing more-than-ample information for
parameter inversion using the double-Manning approach (Jones and Wickert, 2023). For the Cannon River, we limit or use
of stage–discharge data to 2002–2012, during which the stage–discharge relationship appears "stable" (further explained in
Section 5.2) (Jones et al., 2023). A consequence of restricting this analysis to this "stable" time period is that it includes few



**Table 2.** Rivers used to test the double-Manning approach. All units are SI. Discharge start and end are for our calculation of mean discharge, and do not imply the loss of the stream gauge.

| River | Minnesota | Cannon | La Dormida |
|---|---|---|---|
| **Gauge** | | | |
| Name | Jordan | Welch | Captación |
| Operator | USGS | USGS | – |
| Number | 05330000 | 05355200 | – |
| Latitude | 44.69301845 | 44.5638559 | $-0.023482$ |
| Longitude | $-93.641902$ | $-92.7321429$ | $-78.016396$ |
| Drainage area [m$^2$] | $42000 \times 10^6$ | $3470 \times 10^6$ | $6.9 \times 10^6$ |
| Mean Discharge [m$^3$ s$^{-1}$] | 150 | 30 | 1.6 |
| Discharge data start | 1934.10.01 | 1991.10.01 | 2019.01.02 |
| Discharge data end | 2021.07.23 | 2019.08.19 | 2023.06.30 |
| Rating-curve data points | 1131 | 86 | 4 |
| Rating-curve data start | 1934.07.12 | 2002.01.08 | 2018.12.31 |
| Rating-curve data end | 2020.10.29 | 2012.01.19 | 2023.06.30 |
| **Measured** | | | |
| Channel width ($b$) [m] | 100 | 45 | 2.6 |
| Valley-bottom width ($B$) [m] | 1100 | 300 | 8.3 |
| Channel depth ($h_\beta$) [m] | 7 | $2.17 \pm 0.33$ | 0.7 |
| Channel slope ($S$) | 0.0001 | 0.0009 | 0.0788 |
| Grain size ($D_{50}$) [m] | $0.25 \times 10^{-3}$ | [a] $39 \times 10^{-3}$ | $123 \times 10^{-3}$ |
| Grain size ($D_{84}$) [m] | $0.35 \times 10^{-3}$ | [a] $55 \times 10^{-3}$ | $180 \times 10^{-3}$ |
| **Inferred from $D$** | | | |
| Manning's $n_{\mathrm{ch}}$ [s m$^{-1/3}$] | – | [a] $\leq 0.030$ | 0.037 |
| **Solved** | | | |
| Manning's $n_{\mathrm{ch}}$ [s m$^{-1/3}$] | 0.034 | 0.025 | 0.038 |
| $k_{\mathrm{fp}}$ [m$^{3-P_{\mathrm{fp}}}$ s$^{-1}$] | 138 | 39.7 | [b] 11.4 |
| $P_{\mathrm{fp}}$ | 1.62 | [c] 5/3 | [d] 5/3 |
| $n_{\mathrm{fp}}$ [s m$^{-1/3}$] | [b,e] 0.079 | [b] 0.061 | [f] 0.14 |
| Stage Offset ($z_b$) [m] | 0.47 | 0.67 | 0.063 |
| Channel depth ($h_\beta$) [m] | 5.8 | 2.1 | [d] 0.7 |
| Channel width ($b$) [m] | [d] 100 | [d] 45 | [d] 2.6 |
| Solution RMSE [m$^3$ s$^{-1}$] | 44.6 | 8.48 | 0.0245 |
| RMSE / Mean Discharge | 0.30 | 0.28 | 0.015 |

[a] Data obtained from bars at the confluence with a steep tributary $\sim$500 m upstream of the gauge. These likely represent a maximum grain size at the gauge itself, and therefore, likely upper limit on the in-channel Manning's $n$.

[b] Computed following Equation 10

[c] Fixed at boundary values during inversion

[d] Fixed by input data

[e] Back-calculated through a fit in which $P_{\mathrm{fp}}$ is fixed at 5/3.

[f] Fixed at estimated value from qualitative field observations

data points for large floods. Therefore, our goal here is to combine field observations of channel geometry with stage–discharge data to build the best possible rating curve.



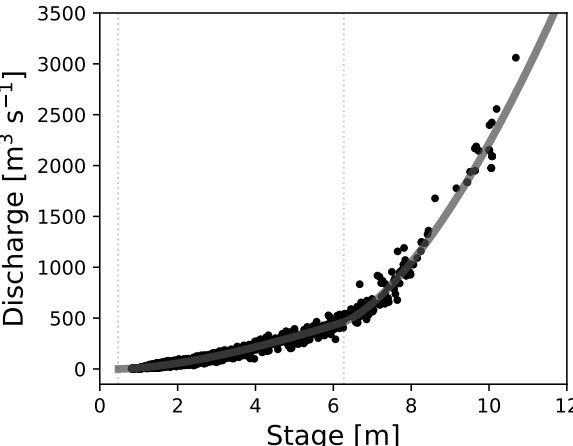

**Figure 2.** Stage–discharge rating curve for the Minnesota River near Jordan. Field measurements of paired stage and discharge appear as black points, and the rating-curve fit overlies them as a semi-transparent gray line. The light gray dotted vertical lines represent (left) the stage at which discharge is 0 (i.e., $z_b$, the parameterized channel-bed elevation) and (right) the bankfull stage at $z_b + h_\beta$. Only nuanced visible differences exist between this best-fitting plot, corresponding to the parameters given in Table 1, and a fit in which we force $P_{\mathrm{fp}} = 5/3$ to calculate a floodplain Manning's $n$. The `doublemanning-fit` software automatically generated this plot and the others like it in this article.

The dataset from remote and poorly accessible La Dormida in the Ecuadorian Andes includes just four discharge measurements (Nelson, 2021; Jacoby et al., 2022; Ng et al., 2023). To demonstrate how extreme data sparsity can be addressed, we

leverage additional observational and theoretical (Gioia and Bombardelli, 2001; Bonetti et al., 2017) constraints in order to apply our double-Manning approach to generate a reasonable rating curve.

### 5.1 Inversion from ample data: Minnesota River

The Minnesota River near Jordan, Minnesota, USA, drains ~42,000 km² of formerly glaciated topography stretching from the eastern portions of the Great Plains to the western edge of the hardwood forests. It meanders across the floor of a valley formed

by the Glacial River Warren (Gran et al., 2013). As a major waterway across a now-agricultural post-glacial landscape, the Minnesota River has been gauged at Jordan since 1934. The combined effects of climate change and agricultural expansion, with associated land drainage, caused (and continues to cause) significant channel widening via rapid delivery of water to the river, resulting in larger floods (Engstrom et al., 2009; Blumentritt et al., 2013; Schottler et al., 2014; Lauer et al., 2017; Kelly, 2019). This has required repeated shifts of the Minnesota River's rating curve since the beginning of gauging in order

to maintain its ability to predict modern stage–discharge relationships. These shifts ensure that the data points relating to stage and discharge correspond to the modern hydraulic geometry.



The 1131 shift-adjusted stage–discharge measurement pairs permit us to apply the double-Manning approach with few constraints from data beyond the stage–discharge points. We prescribe modern river width at $b = 100m$ and slope at $S = 10^{-4}$ (Libby, 2018; Minnesota Department of Natural Resources, 2014). We prescribe width to be a set value because it is easy to
observe and co-varies with Manning's $n$ ($n_{\text{ch}}$), which is more difficult to observe (Table 1; Section 3.2). In-channel Manning's $n$ ($n_{\text{ch}}$) varies between 0.03 and 0.055 at Chaska (U.S. Army Corps of Engineers, 1952), ∼12 km downvalley of the Jordan gauging station. To ensure that we do not overconstrain the problem, we extend our Manning's $n$ search range to $0.25 \leq n_{\text{ch}} \leq 0.60$. We permit channel depth, $h_\beta$, to range between 4 and 10 meters (Kelly, 2019). We leave $k_{\text{fp}}$ and $P_{\text{fp}}$ effectively unconstrained by giving them wide estimation bounds.

The solved parameters (Table 1) lie within expectations from measurements, and the plotted fit (Figure 2) visually appears accurate; see Table 2 for the root-mean-square error (RMSE). $n_{\text{ch}}$ falls within the Chaska data range (U.S. Army Corps of Engineers, 1952), as well as within the $n = 0.020$–$0.040$ range expected for sand-bed rivers with dune-covered beds (Simons et al., 2004). Channel depth ($h_\beta$) is a little less than our field-estimated ∼7 m, but remains close at 5.8 m.

The best-fit $P_{\text{fp}} = 1.62$ is close to $5/3$, which would be expected for a simple full-floodplain inundation without complex
topography or roughness structure. From this finding, we infer that the wide-rectangular floodplain approximation (Section 2.2.1) applies. Such a result is sensible for the Minnesota River, whose steep valley walls (a result of deglacial megaflood incision: Belmont et al., 2011) confine a broad and essentially flat floodplain.

Noting that $P_{\text{fp}}$ corresponds to a near-rectangular floodplain and incorporating our knowledge of valley-bottom width (Table 2), we back-calculate an expected floodplain pseudo-Manning's $n$ from Equation 10 to be 0.072. We refer to this value as
a "pseudo-Manning's $n$" because we compute it with the best-fit $k_{\text{fp}}$ from when $P_{\text{fp}} \neq 5/3$. Re-running the inversion problem and holding $P_{\text{fp}}$ at $5/3$ results in $k_{\text{fp}} = 126$ and a value of $n_{\text{fp}} = 0.079$ for Manning's $n$ itself. Here, floodplains of the Minnesota River comprise forests, fields, and wetlands. Acrement and Schneider (1989) provides Manning's $n$ values of 0.10–0.15 for similarly forested floodplains. Phillips and Tadayon (2006) notes that Manning's $n$ ranges from 0.025–0.050 across floodplain pastures. For submerged vegetation, which we take to be representative of wetlands, Manning's $n < 0.025$ (Phillips
and Tadayon, 2006). These values bracket our computed Manning's $n$ on this mixed-land-cover floodplain and provide some confidence in our results.

Taken together, fitting this Minnesota River stage–discharge data set (Jones and Wickert, 2023) demonstrates two concepts to carry forward. First, *a priori* knowledge of floodplain geometry and/or roughness may aid in pre-specifying $k_{\text{fp}}$ and/or $P_{\text{fp}}$, thereby further constraining special cases of the the stage–discharge problem. Second, the $2.3\times$ difference between $n_{\text{ch}}$
and $n_{\text{fp}}$ indicates the importance of explicitly resolving channel–floodplain differences within the rating-curve fit. In a typical power-law rating-curve approach (Eq. 1), such a sudden change in the coefficient may be absorbed by the exponent. This may not be able to match the bend in the stage–discharge curve associated with a transition to floodplain flow. Furthermore, it will not as effectively encapsulate a physically based parameterization of flow processes (see Gioia and Bombardelli, 2001; Bonetti et al., 2017), thereby making the meaning behind the values of the rating-curve coefficient and exponent harder to interpret. In
contrast, the double-Manning approach permits intuitive adjustments if hydraulic geometry and/or roughness change.




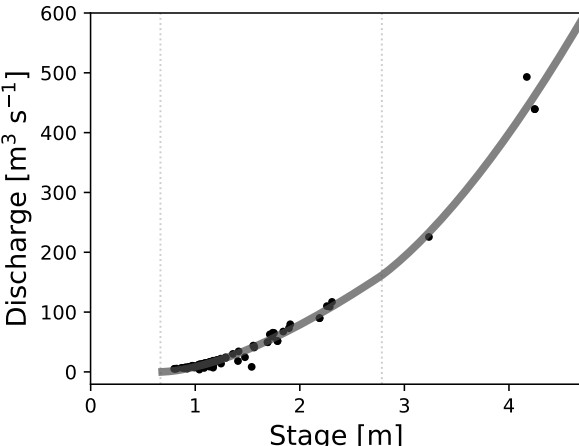

**Figure 3.** Stage–discharge rating curve for the Cannon River at Welch. See explanation of symbols in Figure (2) caption. Though only few data points constrain the bankfull height of the channel, we used the surveyed stream-gauge elevation alongside lidar topographic data and (iteratively) the computed stage offset to constrain our $h_\beta$ search window to 2.17±0.33 m.

## 5.2 Fitting more limited data: Cannon River

The Cannon River in southeastern Minnesota, USA, drains 3470 km$^2$ of woodlands, farms, cities, and forests en route to the Mississippi River valley. Its tributaries cross low-relief uplands, the large western portion of which were ice-covered at the Last Glacial Maximum (Patterson and Hobbs, 1995). Its primary gauge, at the town of Welch, records streamflow within the bedrock-walled and gravel-bedded lower valley of the Cannon River.

When plotting the USGS record of Cannon River stage and discharge, we noted systematic sub-parallel trends, prominent at low-magnitude discharges, that vanished when we examined a shorter time window. These likely indicate changes in channel hydraulic geometry, yet unlike for the Minnesota River data set above, these were not fully corrected via shift adjustments to the rating-curve data. As a result, we generate a rating curve appropriate for a subset of data obtained from 2002–2012. This scenario requires us to reduce the data used, and therefore illustrates how the double-Manning approach may help to fill in gaps within the resultant sparser data set.

As input to `doublemanning-fit`, we measured the hydraulic geometry of the Cannon River (Table 1). Using overhead imagery via Google Earth from the 2002–2012 period, we estimated channel width at the Welch gauge. We computed the slope of the Cannon River from the site of the gauge at Welch to the mouth of Belle Creek, ∼3.5 km downstream, using elevation differences from the Minnesota statewide lidar digital elevation model (DEM) collected in 2011 (Minnesota Department of Natural Resources, 2014). To estimate bank height ($h_\beta$), we found and then combined floodplain-surface elevation with channel-bed elevation. We obtained floodplain-surface elevation ($Z_{fp}$, with the capital $Z$ indicating that this is absolute elevation above sea level) from the Minnesota statewide lidar DEM, with an approximate absolute accuracy of ±0.15 m. We also



observed the spatial variability in floodplain-surface elevation to be approximately $\pm 0.15$ m. To find the channel-bed elevation,
we first obtained the surveyed gauge datum elevation ($Z_g$, accurate to $\pm 0.03$ m). Noting that

$$h_\beta = Z_{\text{fp}} - (Z_g + z_b) \tag{15}$$

(with $Z_g + z_b$ providing the channel-bed elevation above sea level), we then sought the stage offset ($z_b$), corresponding to the
channel-bed elevation with respect to the datum-defining stage. We did this by initially guessing that $h_\beta = Z_{\text{fp}} - Z_g$, computing
$z_b$ through `doublemanning-fit`, and then updating $h_\beta$ using Equation 15. After solving for $z_b = 0.67$ m, we summed error
estimates to constrain $h_\beta$ within the range of $2.17 \pm 0.33$ m. We input this value to `doublemanning-fit` via its YAML file
as our initial parameter estimate (Jones et al., 2023).

The data set contains only three observations of overbank flows, and these constrain $k_{\text{fp}}$ and $P_{\text{fp}}$ only loosely. Because two
of these data points lie close to one another, the data emulate a scenario with two data points, thereby allowing $k_{\text{fp}}$ and $P_{\text{fp}}$ to
trade off against one another in a wide range of parameter pairs that fit the data. This motivates an additional external constraint
to reduce the free-parameter space. Here, we follow our prior advice (Section 3) and set the minimum likely value for $P_{\text{fp}}$ at
5/3.

We then computed the Cannon River stage–discharge rating curve from its 2002–2012 data set (Figure 3) Our inversion
solution returns $P_{\text{fp}} = 5/3$, its prescribed minimum value. We typically would consider a solution at one of the bounding limits
to be a poor result, but here simply accept this based on the lack of available data alongside the good visual and quantitative fit
(Figure 3).

As independent control on channel roughness, and hence, Manning's $n_{\text{ch}}$, we used grain-size data obtained near the Welch
gauge (Jones et al., 2023). $D_{84} = 55$ mm (Table 2), corresponding to a Manning's $n$ value of 0.030 (Equation 14). Based on
access limitations, these data were gathered upstream of the gauge, at the confluence with a steeper tributary. Therefore, they
represent a likely upper bound on the grain-induced in-channel roughness. Our inversion generated $n_{\text{ch}} = 0.025$, consistent
with this bound placed on it from grain-size observations.

Because $P_{\text{fp}} = 5/3$, we once more can compute $n_{\text{fp}}$. The surrounding landscape comprises farmland and scattered forests.
Expected Manning's $n$ values for agricultural fields range from 0.03 (bare) to 0.05 (mature crops) (Phillips and Tadayon, 2006).
Those for roughly similar forests in Mississippi, USA, range from 0.10–0.18. Solving Equation 10 with the estimated $k_{\text{fp}}$, we
find that $n_{\text{fp}} = 0.061$, which lies between the values for the two land-use types.

Analyzing the Cannon River data set teaches us two lessons. First, field-based parameter estimates – in this case, relating to
channel depth and Manning's $n$ – can help to constrain and validate rating-curve parameters. Second, realistic physical bounds
on floodplain hypsometry – and hence, $P_{\text{fp}}$ – may be required where data are sparse. Both this example and that from the
Minnesota River demonstrate that a rectangular floodplain approximation could apply generally.

### 5.3 Fitting and extrapolating: La Dormida

The La Dormida River originates at the tongue of the Glaciar Hermoso (elevation ca. 4680 m) and drains a narrow watershed
on the southwestern flank of Volcán Cayambe, Ecuador. Streamflow in La Dormida comes from a combination of glacier melt,



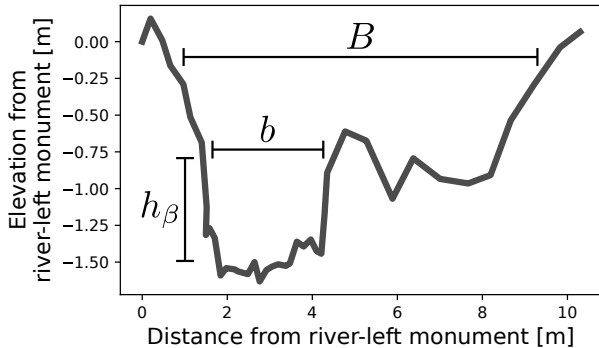

**Figure 4.** Surveyed cross section elevation profile across La Dormida at the Captación gauging site (Ng et al., 2023). The cross section is plotted from river left to river right (i.e., looking downstream). The approximately rectangular channel on valley left with width $b$ and depth $h_\beta$ is inset into a roughly rectangular floodplain of width $B$. The floodplain channel on valley right helps to define the bankfull channel depth. Values for $b$, $h_\beta$, and $B$ were selected visually (Table 2); other approximations are possible.

precipitation, and groundwater (including localized hydrothermal) springs (Nelson, 2021). The stream gauge at La Dormida Captación (elevation 3756 m), comprises a pressure transducer in a stilling well and a mounted staff gauge (Ng et al., 2023). At the gauging site, La Dormida has an extremely steep cobble- to boulder-bed channel: a typical gravel-bed channel has a

slope near 0.01, whereas for La Dormida Captación, $S = 0.0788$. Because of its remote location and development as part of a water-resources research project, only four paired stage–discharge measurements have been made, none of which incorporate overbank flow.

A surveyed cross section at La Dormida Captación (Figure 4) presents an approximately rectangular river channel. This shape validates our rectangular channel approximation for the in-channel portion of this double-Manning approach. Critically,

it also gives us an approximation of $h_\beta = 0.7$ m, which we base on the vertical distance between the channel bed and a height ~10 cm above the surveyed floodplain channel but significantly below the natural levee. This narrow and steep-walled channel also presents a case in which using the hydraulic radius produces a significantly more accurate solution than using the depth via the wide-channel approximation (Equation 7).

Although floodplain topography is more complex than that of the channel, we also approximate its hydraulic geometry to

be rectangular and note that it is ca. $10\times$ wider than it is deep (Figure 4). Therefore, we compute $Q_{\text{fp}}$ using Equation 9. This approach fixes $P_{\text{fp}} = 5/3$, and replaces the lumped $k_{\text{fp}}$ parameter with a field-approximated $n_{\text{fp}}$ and measured valley-bottom width ($B$) (Equation 10). Near the gauge, floodplain vegetation comprises dense coverage by small trees (*Polylepis*), shrubs (e.g., *Hypericum*, *Brachyotum*, and *Escallonia*), and tussock grasses (*Calamagrostis* and *Cortaderia*). By visually comparing floodplain-vegetation character and density to the reference sites from Acrement and Schneider (1989), we assign $n_{\text{fp}} = 0.14$.

This high Manning's $n$ value for the floodplain reduces the relative significance of valley-wall drag compared to flow resistance across the floodplain, further justifying the wide-rectangular floodplain approximation (Section 2.2.1). Combining this $n_{\text{fp}}$ with our estimated valley-bottom width (Figure 4), we use Equation 10 to solve for $k_{\text{fp}}$.



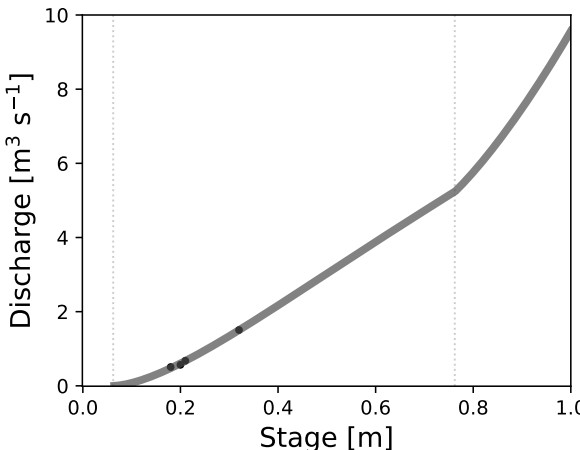

**Figure 5.** Stage–discharge rating curve for La Dormida Captación.

To address the extreme sparsity of stage–discharge data points, we independently infer $n_{\mathrm{ch}}$ from the observed in-channel grain-size distribution (Table 2). From the measured $D_{84} = 180$ mm, we compute $n_{\mathrm{ch}} = 0.037$. We could impose this on the

fit, but instead allow $n_{\mathrm{ch}}$ to remain a free parameter. We then evaluate the best-fitting solution by comparing the computed $n_{\mathrm{ch}}$ against this grain-size-inferred value.

After entering the field-observed channel geometry ($b$, $h_\beta$), floodplain width ($B$), and floodplain roughness ($n_{\mathrm{fp}}$) into equation 9, we inverted the limited stage–discharge data to obtain a best-fitting stage–discharge curve (Figure 5). The estimated $n_{\mathrm{ch}} = 0.38$, which is virtually identical to the $n_{\mathrm{ch}} = 0.37$ value inferred from grain size (Table 2). Although the stage–discharge

relationship for overbank flows remains unconstrained by stage–discharge data, the surveyed hydraulic geometry (Figure 4) and land-cover character provide a physically based first estimate for this portion of the rating curve.

## 6   Discussion

The physical basis of the double-Manning approach offers three notable benefits for establishing river stage–discharge relationships. First, it explicitly distinguishes in-channel and overbank (floodplain) flow, providing the expected inflection in the

rating curve when flows overtop the channel margins. Second, it permits links to and tests against field data that (a) augment the standard paired stage–discharge measurements and (b) in many cases are easier to measure. Third, many of these physically based parameters may continue to be used following changes in hydraulic geometry, which may occur due to individual floods (Hofmeister et al., 2023) or in response to progressive changes in hydrological forcings as climate and land use change (Schottler et al., 2014). We next discuss how this flexibility may enable us to simulate the hydrogeomoprphic coupling among

changing hydraulic geometry, discharge, and shear stress.





## 6.1 Explicit overbank region

Most rivers on Earth self-organize into either a channel and floodplain or – in the case of rapidly incising streams – a channel surrounded by valley walls of a different geometry (Pfeiffer et al., 2017; Naito and Parker, 2019; Turowski et al., 2023). The double-Manning approach acknowledges the natural break that occurs between the in- and beyond-channel regions. This allows

physically meaningful parameters regarding the shape and roughness of these regions.

Making this overbank region and its associated parameters explicit and distinct enables the other two noted advantages of the double-Manning approach. First, it permits direct comparison against and/or guidance from field data, which naturally vary across geomorphic process-domain boundaries (i.e.: channel to floodplain or hillslope: Section 3; Table 1). Second, it separates the effects of channel–overbank-region (often, channel–floodplain) form from the power-law exponent (Equation 1)

and separates channel and overbank-region flow resistance. Therefore, we distinguish roughness and geometric parameters from one another, and can, for example, solve for evolving hydraulic geometry using known roughness values.

## 6.2 Model suitability

We constructed the double-Manning model to streamline rating-curve development for the user while explicitly physically approximating channel–floodplain hydraulics. We hope that the former enables straightforward uptake of the double-Manning

approach, including further implementation and tests of its usefulness. We expect that the latter will improve data–model integration and the predictive capacity of stage–discharge rating curves, including extrapolation to changing hydraulic geometry and/or roughness (Section 6.3).

Although the double-Manning formulation involves seven parameters (Table 1) – many more than the three employed by the standard power-law fit (Equation 1) – we argue that it better represents the physical system and therefore is a more effective

model. Whereas the parameters in the standard power-law fit must be empirically determined, the double-Manning approach separates the constituents of these lumped parameters into physically based and understandable terms. The double-Manning approach thus brings multiple additional and often easily measurable pieces of information to bear on the problem of relating river discharge to water level, thereby enhancing our ability to develop rating curves.

The two least-measurable terms within the double-Manning formulation are $k_{\mathrm{fp}}$ and $P_{\mathrm{fp}}$. However, these retain more physical

meaning than the corresponding $k$ and $P$ of the traditional power-law fit (Equation 1) because they apply only to flows across the floodplain. This floodplain-only definition bounds likely values for $P_{\mathrm{fp}}$. Indeed, our examples (Section 5) demonstrate that floodplain geometry may be approximately rectangular, yielding $P_{\mathrm{fp}} = 5/3$ and an easily defined true "double-Manning" expression. In this case, $k_{\mathrm{fp}}$ scales inversely with $n_{\mathrm{fp}}$ (Equation 8), which may be estimated through field observations (see Acrement and Schneider, 1989).

The simplicity but physical basis of the double-Manning approach, combined with its ready-to-use numerical implementation (Wickert, 2023), establish it as "useful". On one side of it lie distributed hydraulic models (e.g., Pizzuto, 1991; Kean and Smith, 2005; McDonald et al., 2005; Quintero et al., 2021), which are accurate but take substantial work and expertise to apply, and therefore are not part of the common approach to developing rating curves (see World Meteorological Organi-



zation, 2010b). To the other side lies the straightforward power law in Equation 1, which may easily be fit to data, but which

lacks extrapolatability and may connect only indirectly to our physical understanding of flow processes (Petersen-Øverleir, 2005). Therefore, we hope for the double-Manning approach to be an immediate and ready-to-use upgrade for operational river monitoring and prediction efforts.

### 6.3 Geomorphic change and shift adjustments

Flow through rivers and engineering interventions may alter river-channel form and hydraulic properties. Streamflow-driven

shear stresses and sediment transport can modify river-channel depth ($h_\beta$) and width ($b$) (Naito and Parker, 2019, 2020). Large floods and river-engineering efforts may bring sediments of a different grain size to the channel (East et al., 2023; Ylla Arbós et al., 2024), thereby modifying $n_{ch}$. Changes in the balance of sediment and water supply to a river can alter its slope, though rivers are typically large enough that this takes hundreds to thousands of years (Mackin, 1948; Wickert and Schildgen, 2019).

Such changes to hydraulic geometry and flow resistance alter the stage–discharge rating curve of the river. To account for this

change, stage–discharge rating curves are often "shift-adjusted" to incorporate changes in hydraulic geometry that affect the stage–discharge relationship between one measuring time and another (Mansanarez et al., 2019; Hofmeister et al., 2023), such as in the Minnesota River example (Section 5.1). These shifts are most commonly accomplished by changing $z_b$ in Equation 1. However, it may be the channel width rather than the bed elevation that changes. In this case, the power-law coefficient, rather than an offset term raised to the power-law exponent, should be modified. Similarly to the effect of channel-width change,

roughness and channel slope should adjust the coefficient, rather than an offset term, in a rating curve (see Equation 6).

The double-Manning approach improves rating-curve modifications by permitting traditional "shift adjustments" to be replaced with more realistic representations of channel and floodplain change. By making rating-curve adjustments geomorphically explicit, rating curves may be updated dynamically and with limited information. For example, $b$ may be adjusted based on remotely observed changes in channel width, while $n_{ch}$ and the floodplain parameters are held constant.

This added realism has immediate practical application. In geomorphically dynamic yet data-sparse settings (e.g., Birgand et al., 2013; Nelson, 2021), rating curves may be updated based on straightforward geomorphic observations (Table 1). Furthermore, knowing the mode of channel adjustment permits better estimates of changing river-channel conveyance capacity, which impacts flood-hazard potential (e.g., Slater et al., 2015; Slater, 2016; Blom et al., 2017; Slater et al., 2019; Ahrendt et al., 2022; Wood, 2023). Therefore, to facilitate further work into hydraulic–geomorphic interactions and their implications,

the lightweight `doublemanning` software module (Wickert, 2023) implements CSDMS-compliant interfaces for model coupling (Peckham et al., 2013; Overeem et al., 2013; Tucker et al., 2022).

### 7 Conclusions

We offer the double-Manning approach as a tool to build stage–discharge rating curves that incorporates simplified channel and floodplain geometry alongside basic mechanics of steady, uniform, open-channel flow. This additional physical realism

comes with little added end-user complexity: the `doublemanning` package generates straightforward curve fits to measured



stage–discharge data, which may be further informed by field and remotely sensed observations of the channel and floodplain. By replacing lumped parameters with physically based ones, the double-Manning approach enables flexible and physically meaningful adjustments as hydraulic geometry evolves. In three examples, we demonstrated how double-Manning inverse-model solutions can be tested against or informed by field observations, as well as how the double-Manning approach may

provide physically based extrapolations, useful when establishing short-term gauging stations in remote settings.

*Code and data availability.* The `doublemanning` software package is available via GitHub (github.com/MNiMORPH/doublemanning) and archived through Zenodo (Wickert, 2023). It may be installed from PyPI using `pip`: https://pypi.org/project/doublemanning/. Data sets and associated double-Manning files may be downloaded from GitHub and Zenodo as follows: Minnesota River near Jordan – https://github.com/MNiMORPH/stage-discharge_Minnesota-Jordan (Jones and Wickert, 2023); Cannon River at Welch – https://github.com/

MNiMORPH/stage-discharge_Cannon-Welch (Jones et al., 2023); La Dormida Captación – https://github.com/MNiMORPH/stage-discharge_LaDormida-Captacion (Ng et al., 2023).

## Appendix A:  Notation

$\alpha_r$  Coefficient inversely relating Manning's $n$ to Nikuradse roughness ($k_s$) ($\approx 8.1$) [–]

$b$  Channel width [m]

$B$  Valley-bottom width [m]

$D_{50}$  Noncohesive sediment median grain size [m]

$D_{84}$  84th percentile sediment grain size [m]

$g$  Acceleration due to gravity ($\approx 9.807$) [m s$^{-2}$]

$h$  Flow depth [m]

$h_\beta$  Bank height [m]

$k$  Generic power-law stage–discharge coefficient [m$^{3-P}$ s$^{-1}$]

$k_{\mathrm{fp}}$  Floodplain discharge coefficient [m$^{3-P_{\mathrm{fp}}}$ s$^{-1}$]

$k_s$  Nikuradse roughness parameter [m]

$n_{\mathrm{ch}}$  Manning's roughness coefficient within the channel [s m$^{-1/3}$]

$n_{\mathrm{fp}}$  Manning's roughness coefficient for the floodplain [s m$^{-1/3}$]

$P$  Generic power-law stage–discharge exponent [–]



$P_{\mathrm{fp}}$ Power-law flow-depth–discharge exponent for flow across the floodplain [–]

$Q$ Water discharge [m$^3$ s$^{-1}$]

$Q_{\mathrm{ch}}$ Water discharge within and above the channel [m$^3$ s$^{-1}$]

$Q_{\mathrm{fp}}$ Water discharge above the floodplain [m$^3$ s$^{-1}$]

$R_h$ Hydraulic radius [m]

$S$ slope of water surface and river bed [–]

$\bar{u}$ Mean flow velocity [m s$^{-1}$]

$z_b$ River-bed elevation as an offset from the stage datum [m]

$z_s$ Stage (elevation) [m]

$Z_{\mathrm{fp}}$ Surveyed floodplain elevation above sea level [m]

$Z_{\mathrm{g}}$ Surveyed stream-gauge datum ($z_s = 0$) elevation above sea level [m]

*Author contributions.* AW developed the idea, wrote the code, computed the rating curves, and wrote the manuscript. JCJ assembled data for the Minnesota and Cannon Rivers, discussed the ideas, and edited the manuscript. GCN installed and managed the La Dormida stream gauge

(including roughness and cross-sectional surveys), discussed the ideas and their implementation, and substantially edited the manuscript.

*Competing interests.* The authors have no competing financial interests.

*Acknowledgements.* Campbell Dunn wrote an early version of the command-line interface for `doublemanning-fit`. Daniel Stanton, Jeff La Frenierre, Shauna Capron, Leah Nelson, and Ally Jacoby monitored and maintained the stream gauge at La Dormida Captación. Mariana Cardenas measured bed-grain size at La Dormida Captación. The USGS freely provides stream-gauge data for the Minnesota and

Cannon Rivers. Kaya Koraleski, Jake Benbow, and Emma Johnson measured bed-material grain size near the Welch gauge on the Cannon River. This material is based upon work supported by the National Science Foundation under Grants No. 1758795, 1944782, and 1759071. ADW additionally received support through a Humboldt-Forschungsstipendium from the Alexander von Humboldt-Stiftung. The manuscript has been improved via comments from two anonymous referees as well as from Thom Bogaard, Theresa Blume, and Alberto Guadagnini.





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
