# Peer review of "Technical Note: A double-Manning approach to compute robust rating curves and hydraulic geometries"

_EGUsphere, 2025_

## Referee Comment (RC1)

**Peer Review:**
**Technical Note: A double-Manning approach to compute robust rating curves and hydraulic geometries**

**1. Summary of the Paper**

The authors introduce the Double-Manning methodology for developing rating curves ($Q = k\,(z_s - z_b)^P$), which utilizes knowledge of the underlying physics of flow in open channels to minimize the need for ad-hoc parameters when regression models are used to fit observations.

The double-Manning approach is closely related to a suite of modern efforts aimed at developing more flexible, physically grounded rating curves. The authors aim to provide a middle ground between purely empirical fits and full hydrodynamic models.

The authors argue that, compared to other recent methods, their developments are innovative in coupling two Manning equations to reflect channel and floodplain contributions to flow – a concept simple in formulation yet powerful in practice. The concept of double manning emphasizes practical adaptability (via open-source implementation and easily interpretable parameters), whereas some other state-of-the-art methods emphasize comprehensive uncertainty quantification or hydrodynamic completeness. Each approach has its strengths: double-Manning excels in simplicity and physical interpretability, Bayesian methods in statistical rigor, and dynamic models in capturing transient behavior. The existence of these parallel developments underlines a converging theme in hydrology: the need for rating curve models that can handle non-standard conditions (evolving channels, limited data, unsteady flows) more robustly than the old static empiricism. In this context, Wickert et al.'s contribution stands out as a practically minded yet scientifically sound method that complements recent advances. It pushes the field toward rating curves that are mechanistically informed and update-ready, which is an important step for improving flood forecasting, stream monitoring, and water resources management under changing environmental conditions.

**2. Relevance and Coverage of Citations**

The authors of this technical note demonstrate a strong awareness of both the foundational and the latest literature in stage–discharge rating curve development and open-channel hydraulic modeling. They explicitly cite classical, seminal works such as Manning's original formulation for flow resistance (Manning, 1891) and Leopold & Maddock's landmark study on hydraulic geometry (1953). The paper also covers recent advances (within ~10 years)

in rating curve methodology and uncertainty quantification. For example, it cites Kiang et al. (2018), a comprehensive comparison of streamflow uncertainty estimation methods (which includes modern rating curve techniques), as well as Hrafnkelsson et al. (2022), who introduced a generalized power-law rating curve using hydrodynamic theory and Bayesian hierarchical modeling. They also reference Le Coz et al. (2014), an influential study that combined hydraulic knowledge with uncertain gaugings in a Bayesian framework (the "BaRatin" method). The authors even refer to Quintero et al. (2021), which describes "synthetic rating curves" generated via hydrologic/hydraulic models for stage-only gauges, illustrating that they have surveyed contemporary innovations in establishing rating curves when direct measurements are limited.

There do not appear to be obvious omissions of critical recent work.

**3. Originality and Publication History**

This article is an original contribution. We find no evidence that the core ideas or results have been previously published in any journal or formal conference proceedings by these same authors. The methodology appears to be an original synthesis rather than a repackaging of the authors' earlier works.

**4. Comparison to Recent Methods and Tools**

The double-Manning approach enters a landscape of active research on improving rating curves, and it shares goals with several recent methods and tools.

The most closely related developments from the last decade include:

**Bayesian/Physical Hybrid Rating Curves (e.g. BaRatin and RUHM):** Compared to these, the double-Manning approach is less computationally intensive and forgoes an explicit Bayesian treatment of uncertainty in its current form. Its innovation lies in using two applications of Manning's equation (for channel and floodplain zones) as a constrained form of a piecewise rating curve, rather than relying on generic power-law segments or full hydrodynamic simulations. However, it currently does not inherently provide probabilistic uncertainty estimates as RUHM or BaRatin do. The trade-off is between ease-of-use and statistical rigor: double-Manning favors a straightforward, deterministic calibration with physically plausible parameters, while methods like RUHM prioritize a full accounting of uncertainties and leverage advanced computation (MCMC or other Bayesian algorithms) to fuse models and data.

**Generalized Power-Law and Theoretical Extensions:** A notable recent contribution is Hrafnkelsson et al. (2022), who generalized the traditional rating curve by deriving the power-law exponent and coefficient from hydrodynamic considerations and fitting a Bayesian hierarchical model. Their approach maintains the familiar power-law form but

links parameters to physical quantities (like channel shape and flow regimes) and pools information across sites via a hierarchical Bayesian structure. The double-Manning approach shares a similar spirit of physically-informed modeling but implements it more directly: instead of modifying the power-law exponent abstractly, it literally employs Manning's equation in two flow domains. This makes double-Manning somewhat more prescriptive – it assumes a rectangular channel cross-section and, optionally, a rectangular floodplain – whereas Hrafnkelsson's framework is more flexible in form (adapting the power-law curve shape through theory). In terms of innovation, double-Manning's two-tier Manning equation is a fresh idea that effectively creates a compound rating curve without an arbitrary breakpoint; its method of using one Manning relation for in-bank flows and another (or a Manning-like power law) for overbank flows is an innovative yet intuitive extension of classical uniform flow theory.

**Dynamic and Non-Stationary Rating Methods:** Another related thread is the development of rating curve methods that account for non-stationary conditions and flow dynamics (beyond the static stage–discharge assumption). For instance, researchers at the USGS have devised a "dynamic rating" approach to capture hysteresis effects during unsteady flows. Domanski et al. (2022) introduced DYNMOD and DYNPOUND, simplified hydrodynamic models derived from the Saint-Venant equations that can compute discharge from stage while accounting for changing energy slope and storage in the channel/floodplain (hysteresis). These methods effectively produce time-varying rating relationships that adjust during a flood wave, which a single static curve cannot do. The double-Manning method is complementary to such approaches: it addresses spatial complexity (channel vs floodplain flow regimes) and long-term morphological changes, rather than short-term unsteady flow dynamics. Double-Manning assumes quasi-steady uniform flow for given stages, so it will not capture hysteresis loops during events (as DYNMOD/DYNPOUND do).

**5. Strengths and Limitations**

**Strengths of the paper:** The proposed methodology would reduce the need for the multiple measurements required in a purely empirical fit of a rating curve. However, this only seems to be the case when all the hypotheses of the double-manning methodology hold, and the authors do not present evidence that this situation is the most common in cross-sections with rating curves around the world.

**Weaknesses and limitations:** The title suggests a level of generality of the application that is not supported by the results and analyses. The title could more explicitly reflect the methodological context and applicable site conditions—specifically, that it is intended for locations with available stage -discharge measurements and supporting field data. Additionally, emphasizing that the approach is a hybrid hydraulic–empirical model for rating curve fitting would enhance clarity and precision.

The evidence that the methodology of double manning rating curves works is very minimal, and there isn't a formal comparison of errors with existing methodologies.

The paper does not provide direct evidence of the methodology's accuracy, as it is applied to two sites with substantial stage–discharge measurements but without quantitative comparison to a reference or "true" rating curve—such as that provided by the USGS. In the third case, where only a few measurements are available, it is not possible to verify the accuracy of the resulting fit, particularly in the floodplain region where no observational data are available.

We cannot find proof that there is an "economy" of data using this approach. We would have expected that the authors would show that a minimum set of observations is needed to obtain the same or less error than a traditional fit of the data.

**6. Figures**

All the figures should be improved.

Figure 1:
- Clarify flow regions. The distinction between $Q_{ch}$ (channel discharge) and $Q_{fp}$ (floodplain discharge) could be enhanced by using colored shading or distinct arrows for each flow component.

- Adding directional flow arrows to illustrate how flow is distributed above the bankfull stage.

Figures 2, 3, and 5:
- Add gridlines to improve readability.

- Update discharge units from "m3s$^{-1}$" to "m$^3$/s".

- Improve axis labels and enhance the visibility and style of dashed lines.

- Increase the size of the observation points and consider using a different color than the model curve for clearer distinction.

- For Figures 2 and 3, include a reference curve (USGS rating curve) for comparison.

Figure 4:
- Add gridlines and a dashed horizontal line to indicate bankfull elevation, reinforcing the concept of overbank flow.

- Improve labels clarity

**6. Specific Recommendations**

Line 11: While the abstract notes that the method "matches ground truth" and "enables predictions," it does not summarize any specific performance metrics or case study outcomes. Include a brief reference to a specific result or performance.

Line 97: The variability of the coefficients $k_{fp}$ and $P_{fp}$, which are influenced by changes in floodplain width and roughness, is not fully addressed in the paper. In real-world settings, floodplains are often heterogeneous and exhibit substantial spatial variability along the river reach.  However, the method appears to assume spatial homogeneity within the floodplain zone, which may oversimplify real-world conditions. In practice, floodplain heterogeneity introduces uncertainty that could affect the accuracy of the fitted rating curve, particularly in the overbank flow regime. How does the proposed method account for this heterogeneity, and how is the resulting uncertainty represented in the fitted rating curve? Given that these parameters directly influence the overbank component of the discharge, a discussion of how spatial variations and their associated uncertainties affect the reliability of $Q_{fp}$ would strengthen the analysis.

Line 125: Section 3 (Data Constraints): The text mixing parameter estimation difficulty, data source types, and model sensitivity can be dense. Reformat Table 1 to include a column for "Parameter Sensitivity" (if known), and break Section 3 into clearer subsections for: Measurable parameters (e.g., b, S, hβ) - Estimated parameters (e.g., $k_{fp}$, $P_{fp}$) -Data-sparse strategies

Line 247: To better demonstrate the applicability of the methodology to data-limited sites, it would be helpful to conduct a set of controlled experiments at a single site using progressively reduced subsets of data. For example, the authors could evaluate model performance using only channel-stage measurements (excluding overbank flow), then with a few measurements spanning both channel and floodplain stages and compare the resulting rating curves to the full dataset fit. Each case could also be compared against a reference curve (e.g., the USGS rating curve) to assess the sensitivity and robustness of the approach under constrained data conditions (calculate some metrics). This would provide valuable insights into the model's behavior and reliability when applied to real-world scenarios with sparse (or none) observations.

Line 369: Replace "$n_{ch}$ = 0.38, which is virtually identical to the $n_{ch}$ = 0.37" by "$n_{ch}$ = 0.038, which is virtually identical to the $n_{ch}$ = 0.0

Line 372: It could be valuable to include a fourth case study where the channel geometry deviates from the rectangular assumption—for example, a compound channel. This would allow the authors to explore the applicability and limitations of the double-Manning methodology under more complex geometric conditions, which are common in natural river systems. Such an example would also help assess the method's flexibility and the potential need for adjustments when applied to non-idealized cross sections. Including this type of case would further strengthen the practical relevance of the approach.

Line 442: The authors should consider expanding the conclusions section, which currently consists of a single paragraph. In addition to summarizing the strengths of the double-Manning approach, the conclusions should also acknowledge the method's limitations. For example, potential sources of uncertainty—such as assumptions of floodplain homogeneity, sensitivity to field-estimated parameters, and the challenges of validating results in data-sparse settings—deserve mention. Including both the advantages and constraints would provide a more balanced and complete perspective and help guide future applications and developments of the method.

**7. Recommendations**

Reject

---

## Referee Comment (RC3)

**Federico Gómez-Delgado**
**Referee comment on the technical note:**

**A double-Manning approach to compute robust rating curves and hydraulic geometries**

By Andrew D. Wickert, Jabari C. Jones, and Gene-Hua Crystal Ng

**General comments**

This paper introduces a "double-Manning" approach to construct physically informed river stage–discharge rating curves, promoting the practical importance of simple methods that maintain physical realism. The approach is based on the separation of in-channel and overbank flows, allowing for flexible and improved calibration and extrapolation of rating curves, even in dynamic or data-sparse river systems. It seeks to bridge empirical power-law methods and more complex hydraulic models, offering a practical tool for hydrologists to build or refine stage-discharge rating curves using accessible field and remote-sensing data, supporting flexible, and better-informed river monitoring and prediction. The double-Manning formulation, implemented as open-source software, facilitates straightforward application while allowing adjustments based on direct measurements, reasonable assumptions or estimates, and/or observable geomorphic changes. By replacing traditional lumped parameters with field-measurable river hydraulic and geometric properties, the approach aims to contribute to enhancing operational monitoring, flood prediction, and hydro-geomorphic research.

While the fundamental concept of modelling in-channel and overbank flows separately using hydraulic principles (e.g., Manning's equation) is not novel (e.g., Ven Te Chow, 1959; Sellin, 1964; Henderson, 1966; Posey, C.J., 1967; Knight, D.W., Shiono, K., 1990; Knight, D.W. and Abril, B., 1996; Smart, 1999; Mietton et al., 2000; Knight et al., 2009; Fenton, 2015; Kiang et al., 2018; Manasanarez et al., 2019; IWA Publishing, 2024) the double-Manning approach presented in this technical note introduces a specific implementation that is distinct in its simplicity and focus on data-sparse environments. In particular, it builds on a recognized foundation of Manning-based rating-curve research (Leonard et al., 2001; Kean and Smith, 2005; Price, 2009; Frontiers in Water, 2023). Existing methods, including Bayesian rating-curve frameworks and compound channel modelling, have explored multi-stage rating curves and the separation of channel and floodplain roughness using Manning's equation to enhance physical interpretability (Le Coz et al., 2014; Pappenberger et al., 2006). However, I find that the specific formulation and implementation presented in this technical note, while building upon this body of research, still contributes meaningfully by combining novel structural modelling, dynamic geomorphic responsiveness, and practical software integration. This represents an advance over previous empirical or single-zone Manning adaptations by providing a ready-to-use, physics-based, dual-zone framework operationalized in open-source tools, facilitating practical adoption by the hydrological community.

High-quality and technically sound hydrological (discharge) observations are recognized as largely lacking at both national and global scales (WMO, 2010; WMO, 2022). The World Meteorological Organization has repeatedly highlighted in its State of Global Water Resources

reports and Hydrological Observing System initiatives the critical need for reliable discharge data to support water management, flood forecasting, and climate adaptation, with many regions facing data scarcity (WMO, 2022). Recent reviews in the scientific literature (Alfieri et al., 2020; Blöschl et al., 2019) similarly underscore the limitations in discharge data availability and the need for innovative yet practical methods to improve monitoring capacities globally. This context provides clear merit to contributions like this technical note, which offers conceptual and practical, easy-to-use tools to address the operational challenges of maximizing the use of available discharge measurements and developing rating-curves easily. The open-source software implementation, makes this Manning-based dual-zone rating-curve tool publicly available, representing a valuable addition to the technical literature and practice of operational hydrology.

Additionally, I find merit in the technical note's provision of diverse solution strategies under different scenarios of data availability, which is particularly useful for practicing hydrologists and researchers, especially in data-poor settings. The extract provided by the authors clarifies the positioning of the double-Manning approach as a pragmatic middle ground: it offers a simpler, operationally accessible alternative to distributed hydraulic models while providing greater physical relevance and extrapolation capability than straightforward empirical power-law fits. This simplicity, paired with its physical basis and ready-to-use numerical implementation, underlines the utility of the approach for operational river monitoring and prediction, aligning well with the needs of agencies and practitioners seeking robust yet practical solutions.

In think the title of this technical note might better reflect the sound approach to developing rating-curves that maximize measured hydrologic data and direct field observations of river hydraulics. Furthermore, the indication that the method computes "hydraulic geometries" is unclear (how about instead saying that it provides estimates of geometric hydraulic parameters?).

This note addresses relevant scientific questions within the scope of HESS by focusing on operational hydrology, river monitoring, and methods to improve stage–discharge rating curves using physically informed, practical approaches for dynamic and data-sparse conditions. It presents a novel combination of concepts, practical tools, and implementation/solution strategies, reaching substantial and applicable conclusions. The methods and assumptions are valid, clearly outlined, and sufficient to support the interpretations and conclusions provided. The description of the numerical implementation and conceptual framework is complete and precise enough to allow reproduction by fellow scientists, ensuring traceability of results. The authors give proper credit to related work while clearly indicating their new contributions, which are explicitly differentiated from existing studies.

The paper is very well written, and while I will provide minor recommendations in my specific and technical comments to further improve the text, figures, and tables, it already presents a concise and complete abstract summarizing the work effectively. The overall presentation is well-structured and clear, with fluent and precise language throughout. Mathematical formulae, symbols, abbreviations, and units are correctly defined and used consistently. The number and quality of references are appropriate and sufficient to support the context and contributions of the

work, and the supplementary material provided is of adequate quality and quantity to complement the technical note without redundancy.

In conclusion, I recommend that the technical note be accepted for final publication, subject to further clarifications and technical corrections.

**Specific comments**

**L.68:** noting that $h$ is flow (i.e., water) depth and $h_\beta$ is the height of the channel banks, and $\wedge$ indicates that the smaller of the two numbers be taken

**L.102:** In the statement "Furthermore, we posit that the inundation width and depth distributions can be described with power-law functions.", what do you mean by the term "distribution"? Is the (frequency?) distribution what you really want to describe with this function? On what grounds do you propose power-law functions for this? (Perhaps you could include some reference(s) here).

**L.106:** In "…and therefore rewrite Equation 8 as…", are you actually rewriting Eq. 8, or are you just directly applying Manning's equation (as you did in Eqs 6 and 7), which has a similar structure to Eq. 8?

**L.108:** since the definition of $B$ (the width of the valley bottom) is relative arbitrary, some recommendations or guides on how it could be determined in the field or by remote sensing, for use within the framework of this methodology, could be of great value and use.

**L.117-118:** In "Therefore, we consider the wide-rectangular floodplain approximation to be reasonable even when not formally defensible based on Equation 3 alone.", could you expand/explain this further, for example by mentioning which principles or assumptions necessary to apply Eq. 3 might not be defensible?

**L.136-137:** In "…Geometric and velocity data are measured directly,…", do you really need velocity to estimate any of the parameters in Table 1?

**L.141-142:** Could you please explain in more detail the statement "Therefore, the approximate "rectangular-channel" values for both width and depth should be selected with flow mechanics in mind (e.g., Naito and Parker, 2019)."?

**L.144-145:** In "Therefore, the effective channel depth for the double-Manning approach will be less than the distance from the floodplain surface to the thalweg.", can you introduce first the concept of "effective channel depth" in your explanation? Since the thalweg is the lowest point of the cross section this necessarily implies the main rectangular channel, however, could you explain why such effective channel depth excludes the floodplain? Finally, can you explain why the effective channel depth is less than the distance from the floodplain surface to the thalweg?

**L.154-155:** Can you further explain the statement "(Although $R_h$ also includes dependence on $b$, most channels are wide relative to their depth, making $R_h$ much more sensitive to $h$ than to $b$.)".

**L.158:** In "Similarly straightforward measurements can provide slope ($S$) for Equation 6. This can be done with digital elevation models,…", again, is a DEM-based estimate good enough, given that $S$ is the channel-bed (not surface) slope?

**L.175-177:** In this statement "Although channel-bank height may be solved for as a free parameter using a large amount of stage–discharge data (Section 5.1),…", if you already have a large amount of stage–discharge observations, why would you want to estimate the channel-bank height? From a practical perspective, you could simply fit an empirical rating curve based on your good-quality observations.

**L.209-210:** "Equation 12 involves four field-measurable parameters: channel width ($b$), bank height ($h_\beta$), channel-bed elevation ($z_b$), and in-channel Manning's n ($n_{ch}$). It also includes two free parameters requiring selection or calibration, the power-law coefficient ($k_{fp}$) and exponent ($P_{fp}$) for flows across the floodplain, which relate to floodplain topography and roughness." I think we could also consider the slope $S$ and valley-bottom width $B$ (present in Eqs 6 and 9, which contribute to Eq. 12) as field-measurable parameters, and $n_{fp}$ (from Eq. 9) as a free parameter requiring selection or calibration. Also, note that in Eqs 8 and 9 the channel-bed elevation is presented as the height of the channel banks $h_\beta$. Using different symbols for the same physical concept can be confusing, so please consider using only one or the other throughout the document.

**L.214:** In "Users can specify values for width ($b$), depth ($h_\beta$), and/or slope ($S$); they may also specify bounds for in-channel Manning's $n$ ($n_{ch}$), the floodplain coefficient ($k_{fp}$) and/or exponent ($P_{fp}$), and the offset between flow depth and river stage ($z_b$).", please consider including $B$ and $n_{fp}$ in this list, in case Eq. 9 is required.

**L.280-281:** I do not think the statement, "These values bracket our computed Manning's $n$ on this mixed-land-cover floodplain and provide some confidence in our results." Is justified for a value of $n_{fp} = 0.079$, especially when compared to the criterion of $n < 0.025$.

**L.303-304:** In "We computed the slope of the Cannon River from the site of the gauge at Welch to the mouth of Belle Creek, ~3.5 km downstream", wouldn't this distance be too great to provide an accurate river channel-bed slope for the gauging site?

**L323.324:** I suggest reviewing the statement "…, but here simply accept this based on the lack of available data alongside the good visual and quantitative fit (Figure 3)." In this situation, I would rather refer to the fact that it is better to have an estimate that provides a good visual and quantitative fit than to rely on a purely theoretical solution.

**L.328-329:** Could you further explain the statement "Therefore, they represent a likely upper bound on the grain-induced in-channel roughness."

**L.375-376:** In "Second, it permits links to and tests against field data that (a) augment the standard paired stage–discharge measurements", what do you mean by " augment"?

**Figure 1:**

- It might be worthwhile to also indicate in the title of this figure that both $z_s$ and $z_b$ are measured with respect to a common datum.

**Table 1:**

- Why is the valley-bottom width $B$ excluded? This is one of the parameters required to apply equation 9.

- I would change the title of the "Variable" column to "Symbol" instead.

- Is using a DEM a valid option to estimate $S$ (channel-bed slope) and $h_\beta$ (bank height), considering that estimating both parameters requires bottom/underwater measurement?

- The description of the observation method for the floodplain discharge coefficient $k_{fp}$ should mention that $n_{fp}$ is first estimated through insight obtained from field surveys or Manning's $n$ tables or photos and then entered into Eq. 9 to calculate $k_{fp}$.

**Table 2:**

- Perhaps the rows on grain size $D_{50}$ and $D_{84}$ could be regrouped under a subsection entitled "Inputs for $n_{ch}$"

- In the "Solved" description of the row about $k_{fp}$, since Eq. 8 is empirical, I don't think either $k_{fp}$ or $P_{fp}$ should be assigned any units (I would remove these [$m^{3-fp}\,s^{-1}$] units).

- It would be helpful if the table clearly distinguished between the values of observed variables/parameters (obtained through direct measurement or field-based estimation) and those estimated or optimized using the "doublemanning" software. One option could be to present observed values in bold, with the corresponding estimated values shown in parentheses and in regular font next to them. This would improve the table's readability and help avoid confusion in rows under the "Solved" section, such as "Channel depth ($h_\beta$)", for which observed values are available, and "Stage Offset ($z_b$)", which section 4.2 of this manuscript identifies as a field-measurable parameter.

- I would not include the row "Channel width ($b$)" under the "Solved" section at all. This would simplify the table and perhaps eliminate footnote $d$ (to be checked).

- I recommend that footnotes $c$ and $d$ be better explained, especially when applied to fixing $P_{fp}$.

**Technical corrections**

Below I recommend technical and typographical corrections to this manuscript, and some typing suggestions.

**L.74**: "Most natural channels and floodplains satisfy this criterion …"

**L.116-117:** "However, many floodplains contain such significant internal bottom roughness (e.g., from vegetation) that the additional drag against their side walls is small in comparison."

**L.136:** "The third factor indicates how directly the parameter  in question may be measured or calculated."

**Eq. 14:** I would present the units of the equation separately, leaving the equation clean as just: $nch \approx 0.049\, D_{84}^{1/6}$. The way it is currently presented is confusing, as the units (s m$^{-1/2}$) appear to be variables or parameters of the equation.

**L.181-182:** "…these in fact amplify the overall trend towards wider flow horizontal occupation as water rises…."

**L.187-188:** "…may be  used to compute the bulk coefficient, $k_{fp}$ (Equation 10)."

**L.209:** "…channel width (b)…"

**L.232:** "We apply this double-Manning approach to three rivers (Table 2) to demonstrate its applicability across a  range of settings and quantities of available data…." (I wouldn't talk about a wide range of settings. A "fair" range of settings, maybe).

**L.262-263:** "To ensure that we do not over constrain the problem, we extend our Manning's n search range to 0.025 ≤ nch ≤ 0.060."

**L.356-357:** "..., and estimates  the lumped $k_{fp}$ parameter using  a field-approximated $n_{fp}$ and measured valley-bottom width (B) and slope (S) (Equation 10)"

**L.361-362:** "Combining this $n_{fp}$ with our estimated valley-bottom width (Figure 4) and slope, we use Equation 10 to solve for $k_{fp}$."

**L.367-368:** "After entering the field-observed channel geometry and slope (b, $h_\beta$, S), floodplain width (B), and floodplain roughness ($n_{fp}$) into equation 9,…"

**L.388-389:** "Second, it separates the effects of channel–overbank-region (often, channel–floodplain) form and flow resistance from the power-law exponent (Equation 1)"

**L.398:** "Although the double-Manning formulation involves eight  parameters (Table 1)"

**L.422-423:** "Changes in the balance of sediment and water supply to a river can alter its slope, though large to medium-sized rivers are typically large enough that this takes hundreds to thousands of years (Mackin, 1948; Wickert and Schildgen, 2019)."

**References**

Alfieri, L., Lorini, V., Hirpa, F. A., and Harrigan, S.: A global streamflow reanalysis for 1980–2018, J. Hydrol., 601, 126–450, https://doi.org/10.1016/j.jhydrol.2021.126450, 2020.

Blöschl, G., Hall, J., Parajka, J., Perdigão, R. A. P., Merz, B., Arheimer, B., and Viglione, A.: Changing climate shifts timing of European floods, Science, 357, 588–590, https://doi.org/10.1126/science.aan2506, 2019.

Chow, V. T.: Open-channel hydraulics. McGraw-Hill, 1959.

Fenton, J. D.: Some aspects of flow resistance in compound channels, J. Hydraul. Res., 53, 11–20, https://doi.org/10.1080/00221686.2014.962092, 2015.

Frontiers in Water: Development and uncertainty analysis of rating curves in mountainous watersheds, Front. Water, 5, 1323139, https://doi.org/10.3389/frwa.2023.1323139, 2023.

Henderson, F. M.: Open Channel Flow. MacMillan Company, 1966.

IWA Publishing: Prediction of the flow resistance in non-prismatic compound channels. Water Practice & Technology, 19(5), 1822–1836, 2024. (This refers to a recent publication summarizing established concepts, including momentum transfer as discussed by Sellin).

Kean, J. W., and Smith, J. D.: Generation and verification of theoretical rating curves in steep channels, Water Resour. Res., 41, W05466, https://doi.org/10.1029/2004WR003799, 2005.

Knight, D.W., Shiono, K.: Turbulence measurements in a shear layer region of a compound channel. J. Hydraul. Res. IAHR, 28(2), 175–196, 1990.

Knight, D.W., Abril, B.: Refined calibration of a depth-averaged model for turbulent flow in a compound channel. Proc. ICE Water Maritime and Energy, 118(3), 151–159, 1996.

Knight, D. W., Omran, M., and Tang, X.: Modelling depth-averaged velocity and boundary shear in rectangular compound channels, J. Hydraul. Eng., 135, 245–256, https://doi.org/10.1061/(ASCE)0733-9429(2009)135:4(245), 2009.

Leonard, L. A., Croft, A. L., and Sylvester, S. M.: The use of modified Manning's equation to estimate tidal marsh flow, J. Hydrol., 246, 62–69, https://doi.org/10.1016/S0022-1694(01)00363-4, 2001.

Pappenberger, F., Beven, K. J., Hunter, N. M., Bates, P. D., Gouweleeuw, B. T., Thielen, J., and De Roo, A. P. J.: Cascading model uncertainty from medium range weather forecasts (10 days) through a rainfall–runoff model to flood inundation predictions within the European Flood Forecasting System (EFFS), Hydrol. Earth Syst. Sci., 10, 373–388, https://doi.org/10.5194/hess-10-373-2006, 2006.

Posey, C.J.: Computation of discharge including over-bank flow. Civil Eng. ASCE, 37(4), 62–63, 1967.

Price, R. K.: Improved method for estimating floodplain roughness coefficients, J. Hydraul. Eng., 135, 381–386, https://doi.org/10.1061/(ASCE)0733-9429(2009)135:5(381), 2009.

Sellin, R. H. J.: A laboratory investigation into the interaction between the flow in the channel and that on the floodplain of a river. La Houille Blanche, (7), 793-802, 1964.

Smart, G. M.: Estimating river discharge from sparse stage measurements, J. Hydrol., 211, 182–190, https://doi.org/10.1016/S0022-1694(99)00130-7, 1999.

World Meteorological Organization (WMO): Manual on Stream Gauging, Volume I, Fieldwork, WMO-No. 1044, Geneva, Switzerland, 2010.

World Meteorological Organization (WMO): State of Global Water Resources 2022, WMO-No. 1333, Geneva, Switzerland, 2022.

---

## Author Comment (AC2)

Peer Review: Technical Note: A Double-Manning approach to compute robust rating curves and hydraulic geometries

*1. Summary of the Paper*

The authors introduce the Double-Manning methodology for developing rating curves (Q = k $(z_s−z_b)$P), which utilizes knowledge of the underlying physics of flow in open channels to minimize the need for ad-hoc parameters when regression models are used to fit observations.

The double-Manning approach is closely related to a suite of modern efforts aimed at developing more flexible, physically grounded rating curves. The authors aim to provide a middle ground between purely empirical fits and full hydrodynamic models.

The authors argue that, compared to other recent methods, their developments are innovative in coupling two Manning equations to reflect channel and floodplain contributions to flow – a concept simple in formulation yet powerful in practice. The concept of double manning emphasizes practical adaptability (via open-source implementation and easily interpretable parameters), whereas some other state-of-the-art methods emphasize comprehensive uncertainty quantification or hydrodynamic completeness. Each approach has its strengths: double-Manning excels in simplicity and physical interpretability, Bayesian methods in statistical rigor, and dynamic models in capturing transient behavior. The existence of these parallel developments underlines a converging theme in hydrology: the need for rating curve models that can handle non-standard conditions (evolving channels, limited data, unsteady flows) more robustly than the old static empiricism. In this context, Wickert et al.'s contribution stands out as a practically minded yet scientifically sound method that complements recent advances. It pushes the field toward rating curves that are mechanistically informed and update-ready, which is an important step for improving flood forecasting, stream monitoring, and water resources management under changing environmental conditions.

*Thank you for the well-written and comprehensive note on one of the two major goals (the straightforward and physically based rating curve). We note that this review focuses exclusively on this goal. Our second major goal is to build an approach that integrates geomorphological research and observations, which can aid efforts to build effective rating curves. We appreciate the comments from this review, and further take this focus as a call to draw further attention to the inclusion of geomorphology in a revision.*

**2. Relevance and Coverage of Citations**

The authors of this technical note demonstrate a strong awareness of both the foundational and the latest literature in stage–discharge rating curve development and open-channel hydraulic modeling. They explicitly cite classical, seminal works such as Manning's original formulation for flow resistance (Manning, 1891) and Leopold & Maddock's landmark study on hydraulic geometry (1953). The paper also covers recent advances (within ~10 years) in rating curve methodology and uncertainty quantification. For example, it cites Kiang et al. (2018), a comprehensive comparison of streamflow uncertainty estimation methods (which includes modern rating curve techniques), as well as Hrafnkelsson et al. (2022), who introduced a generalized power-law rating curve using hydrodynamic theory and Bayesian hierarchical modeling. They also reference Le Coz et al. (2014), an influential study that combined hydraulic knowledge with uncertain gaugings in a Bayesian framework (the "BaRatin" method). The authors even refer to Quintero et al. (2021), which describes "synthetic rating curves" generated via hydrologic/hydraulic models for stage-only gauges, illustrating that they have surveyed contemporary innovations in establishing rating curves when direct measurements are limited.

There do not appear to be obvious omissions of critical recent work.

*We are glad to read that the citations seem complete. Thank you for the thorough review.*

**3. Originality and Publication History**

This article is an original contribution. We find no evidence that the core ideas or results have been previously published in any journal or formal conference proceedings by these same authors. The methodology appears to be an original synthesis rather than a repackaging of the authors' earlier works.

*True; thank you for checking.*

**4. Comparison to Recent Methods and Tools**
The double-Manning approach enters a landscape of active research on improving rating curves, and it shares goals with several recent methods and tools.

The most closely related developments from the last decade include:

**Bayesian/Physical Hybrid Rating Curves (e.g. BaRatin and RUHM):** Compared to these, the double-Manning approach is less computationally intensive and forgoes an

explicit Bayesian treatment of uncertainty in its current form. Its innovation lies in using two applications of Manning's equation (for channel and floodplain zones) as a constrained form of a piecewise rating curve, rather than relying on generic power-law segments or full hydrodynamic simulations. However, it currently does not inherently provide probabilistic uncertainty estimates as RUHM or BaRatin do. The trade-off is between ease-of-use and statistical rigor: double-Manning favors a straightforward, deterministic calibration with physically plausible parameters, while methods like RUHM prioritize a full accounting of uncertainties and leverage advanced computation (MCMC or other Bayesian algorithms) to fuse models and data.

*We agree.*

**Generalized Power-Law and Theoretical Extensions:** A notable recent contribution is Hrafnkelsson et al. (2022), who generalized the traditional rating curve by deriving the power-law exponent and coefficient from hydrodynamic considerations and fitting a Bayesian hierarchical model. Their approach maintains the familiar power-law form but links parameters to physical quantities (like channel shape and flow regimes) and pools information across sites via a hierarchical Bayesian structure. The double-Manning approach shares a similar spirit of physically-informed modeling but implements it more directly: instead of modifying the power-law exponent abstractly, it literally employs Manning's equation in two flow domains. This makes double-Manning somewhat more prescriptive – it assumes a rectangular channel cross-section and, optionally, a rectangular floodplain – whereas Hrafnkelsson's framework is more flexible in form (adapting the power-law curve shape through theory). In terms of innovation, double-Manning's two-tier Manning equation is a fresh idea that effectively creates a compound rating curve without an arbitrary breakpoint; its method of using one Manning relation for in-bank flows and another (or a Manning-like power law) for overbank flows is an innovative yet intuitive extension of classical uniform flow theory.

*Thank you for the compliment to our approach. I would like to add that I found the Hrafnkelsson et al. approach to be clever and well-reasoned. Our goal in developing the double-Manning approach is to overcome the common assumption of flow through a concave wetted perimeter (Hrafnkelsson et al., Fig. 2). Their example cases do follow such a geometry (Hrafnkelsson et al., Fig. 4). This is where a functional form that includes the possibility – though not requirement – of including a floodplain becomes an important piece of reality. This is what we provide in the double-Manning approach, and this is what a different f(h) could enable within the Hrafnkelsson et al. (2022) approach.*

**Dynamic and Non-Stationary Rating Methods:** Another related thread is the development of rating curve methods that account for non-stationary conditions and flow

dynamics (beyond the static stage–discharge assumption). For instance, researchers at the USGS have devised a "dynamic rating" approach to capture hysteresis effects during unsteady flows. Domanski et al. (2022) introduced DYNMOD and DYNPOUND, simplified hydrodynamic models derived from the Saint-Venant equations that can compute discharge from stage while accounting for changing energy slope and storage in the channel/floodplain (hysteresis). These methods effectively produce time-varying rating relationships that adjust during a flood wave, which a single static curve cannot do. The double-Manning method is complementary to such approaches: it addresses spatial complexity (channel vs floodplain flow regimes) and long-term morphological changes, rather than short-term unsteady flow dynamics. Double-Manning assumes quasi-steady uniform flow for given stages, so it will not capture hysteresis loops during events (as DYNMOD/DYNPOUND do).

*This is a good note. The double-Manning approach includes the implicit assumption that body forces of the water and channel slope are balanced by frictional resistance along the channel margins, without pressure-force terms.*

**5. Strengths and Limitations**

**Strengths of the paper:** The proposed methodology would reduce the need for the multiple measurements required in a purely empirical fit of a rating curve. However, this only seems to be the case when all the hypotheses of the double-manning methodology hold, and the authors do not present evidence that this situation is the most common in cross-sections with rating curves around the world.

*Thank you for this note and opportunity to explain the rectangular-chan. Rating-curve approaches are often developed with a range of open-channel geometries, including triangular, rectangular, and parabolic channels, as well as "compound" geometries with channels inset within channels. Natural alluvial river channels have forms that are widely approximated to be rectangular – so much so that compilations of channel geometry simply include bankfull width and depth (Trampush et al., 2014). Conditions for bank stability and equilibrium hydraulic geometry generally lead to these rectangular channel forms (see, e.g., Parker, 1978; Dunne & Jerolmack, 2020) with wide aspect ratios (Trampush et al., 2014).*

*Based on this comment, we will clarify the basis for the rectangular channel assumption and its widespread applicability in nature. It is not perfect, but can suffice for a straightforward fit to data with minimal geometric constraints.*

**Weaknesses and limitations:** The title suggests a level of generality of the application that is not supported by the results and analyses. The title could more explicitly reflect the methodological context and applicable site conditions—specifically, that it is intended for locations with available stage -discharge measurements and supporting field data.

*All methods for developing rating curves require some amount of field data, be they stage–discharge data and/or geometries in the field. Therefore, our impression is that adding this information to the title would make it long and include information that readers would believe to be implicit.*

Additionally, emphasizing that the approach is a hybrid hydraulic–empirical model for rating curve fitting would enhance clarity and precision.

*This is a good point, especially because we thought that we had made this clear. The "hydraulic–empirical" language is good, and we will seek to use it in a revision.*

The evidence that the methodology of double manning rating curves works is very minimal, and there isn't a formal comparison of errors with existing methodologies. The paper does not provide direct evidence of the methodology's accuracy, as it is applied to two sites with substantial stage–discharge measurements but without quantitative comparison to a reference or "true" rating curve—such as that provided by the USGS. In the third case, where only a few measurements are available, it is not possible to verify the accuracy of the resulting fit, particularly in the floodplain region where no observational data are available.

*As a Technical Note, the purpose of this article is to present a "new development" of "methods and techniques". It should be "a few pages only" and already significantly exceeds this. We provided three examples of cases in which the double-Manning approach worked. We provide RMSE values as goodness-of-fit estimates. Comparisons to established rating curves, while helpful when considering implementation, would enter the scope of model intercomparison and go beyond presentation of a new technique. These seem to us to be sufficient to show that the method has promise, which is not a comprehensive test or intercomparison as the reviewer requests, but we believe this to be within scope of the goals of the article type.*

We cannot find proof that there is an "economy" of data using this approach. We would have expected that the authors would show that a minimum set of observations is needed to obtain the same or less error than a traditional fit of the data.

*Thank you for the chance to clarify. A traditional fit to data would not have yielded the break in the stage–discharge relationship between in-channel and on-floodplain flows, which we demonstrate in the example from La Dormida. This is because stage–discharge observations exist only for conditions in which the flow is confined by the channel banks. Therefore, the double-Manning approach can provide a physically grounded extrapolation.*

*Regarding the note on "economy" of data, this is not a claim that we make.*

**6. Figures**

All the figures should be improved.

Figure 1:

        Clarify flow regions. The distinction between $Q_{ch}$ (channel discharge) and $Q_{fp}$ (floodplain discharge) could be enhanced by using colored shading or distinct arrows for each flow component.

        Adding directional flow arrows to illustrate how flow is distributed above the bankfull stage.

Figures 2, 3, and 5:

        Add gridlines to improve readability.

        Update discharge units from "m3s⁻¹" to "m³/s".

        Improve axis labels and enhance the visibility and style of dashed lines.

        Increase the size of the observation points and consider using a different color than the model curve for clearer distinction.

        For Figures 2 and 3, include a reference curve (USGS rating curve) for comparison.

Figure 4:

        Add gridlines and a dashed horizontal line to indicate bankfull elevation, reinforcing the concept of overbank flow.

        Improve labels clarity

*We thank you for your recommendations and appreciate your attention to our figures. We will take those suggestions for Figure 1 into consideration, and specifically will seek to make it clearer that the demonstrated flow is above bankfull. On Figure 4, the labels are quite large; we do not know how they can be made clearer. On Figures 2, 3, and 5, the data points are a different color than the model curve. As noted below, we consider model intercomparison to be beyond the scope of this Technical Note. Other changes are stylistic and we will consider them.*

**6. Specific Recommendations**

Line 11: While the abstract notes that the method "matches ground truth" and "enables predictions," it does not summarize any specific performance metrics or case study outcomes. Include a brief reference to a specific result or performance.

*We will update the abstract to be more specific about the data that are matched and include the relevant performance metric.*

Line 97: The variability of the coefficients $k_{fp}$ and $P_{fp}$, which are influenced by changes in floodplain width and roughness, is not fully addressed in the paper. In real-world settings, floodplains are often heterogeneous and exhibit substantial spatial variability along the river reach. However, the method appears to assume spatial homogeneity within the floodplain zone, which may oversimplify real-world conditions. In practice, floodplain heterogeneity introduces uncertainty that could affect the accuracy of the fitted rating curve, particularly in the overbank flow regime. How does the proposed method account for this heterogeneity, and how is the resulting uncertainty represented in the fitted rating curve? Given that these parameters directly influence the overbank component of the discharge, a discussion of how spatial variations and their associated uncertainties affect the reliability of $Q_{fp}$ would strengthen the analysis.

*Contrary to the observation of the referee, $k_{fp}$ and $P_{fp}$ allow us to simulate heterogeneous floodplains. Variability in floodplain topography affect its hydraulic radius in a way that become a function of flow depth. This approach using $P_{fp}$ indeed emulates the generic power-law function for a rating curve. We note this in Lines 95–99. We will consider ways of expanding and clarifying this that do not significantly increase the length of this Technical Note.*

Line 125: Section 3 (Data Constraints): The text mixing parameter estimation difficulty, data source types, and model sensitivity can be dense. Reformat Table 1 to include a column for "Parameter Sensitivity" (if known), and break Section 3 into clearer subsections for: Measurable parameters (e.g., b, S, hβ) - Estimated parameters (e.g., $k_{fp}$, $P_{fp}$) -Data-sparse strategies

*Thank you for this suggestion. We will consider it when revising the paper.*

Line 247: To better demonstrate the applicability of the methodology to data-limited sites, it would be helpful to conduct a set of controlled experiments at a single site using progressively reduced subsets of data. For example, the authors could evaluate model performance using only channel-stage measurements (excluding overbank flow), then

with a few measurements spanning both channel and floodplain stages and compare the resulting rating curves to the full dataset fit. Each case could also be compared against a reference curve (e.g., the USGS rating curve) to assess the sensitivity and robustness of the approach under constrained data conditions (calculate some metrics). This would provide valuable insights into the model's behavior and reliability when applied to real-world scenarios with sparse (or none) observations.

*This suggestion seems like a possibility for a follow-on study. The purpose of this Technical Note, which is necessarily limited in length and scope, is to present and demonstrate a new method to develop rating curves that takes advantage of common features of channel and floodplain morphology.*

Line 369: Replace "$n_{ch}$ = 0.38, which is virtually identical to the $n_{ch}$ = 0.37" by "$n_{ch}$ = 0.038, which is virtually identical to the $n_{ch}$ = 0.0

*Thank you for noting this typo. Indeed, the Manning's n was 0.038, which is virtually identical to the 0.037 value computed inferred from the bed-material grain-size distribution. This will be corrected.*

Line 372: It could be valuable to include a fourth case study where the channel geometry deviates from the rectangular assumption—for example, a compound channel. This would allow the authors to explore the applicability and limitations of the double-Manning methodology under more complex geometric conditions, which are common in natural river systems. Such an example would also help assess the method's flexibility and the potential need for adjustments when applied to non-idealized cross sections. Including this type of case would further strengthen the practical relevance of the approach.

*The authors contest that the "compound channel", which is a channel inset within a broader channel, while common in the rating-curve literature, is not a significant consideration in most natural river systems. Natural rivers tend to have a rectangular to trapezoidal channel geometry. A rectangular channel occurs predominantly in straight reaches (Parker et al., 1978), which are commonly associated with stream-gauging sites. Such reaches could be more closely approximated as trapezoidal, though the broad aspect ratio of most alluvial river channels (i.e., most river channels on Earth) means that the use of a trapezoidal approximation will not significantly change the result. A trapezoidal channel may also be used if one wishes to more appropriately characterize flows moving around a bend (e.g., Dietrich et al., 1979), in which case the point bar can be a long limb of the trapezoid. To answer possible questions of curiosity, there are dynamic reasons for the sparsity of natural "compound" channels, and these generally relate to the fact that shallow regions on the edges of rivers (a) require symmetrical*

*erosion and then deposition to form, which is unusual in curved channels (i.e., many in nature), and (b) that these shallow regions, if they do form, will receive a net flux of sediment from the channel center, causing them to fill. However, such  are beyond the scope of this manuscript, and especially a short Technical Note.*

*As a result of this, cases of "compound channels" in the literature often use the "compound channel" approach to simulate flow across a channel and a local floodplain (e.g., Carling et al., 2002). Indeed, some studies define the "compound channel" as a channel + floodplain (Myers & Brennan, 1990; Liu et al., 2025). This situation is something for which the double-Manning approach is well suited.*

*Therefore, it seems that the manuscript already includes compound channels, at least insofar as their main purpose exists, while allowing irregular floodplain geometries (topographic highs and lows) and different roughness on the floodplains than in the channel.*

Line 442: The authors should consider expanding the conclusions section, which currently consists of a single paragraph. In addition to summarizing the strengths of the double-Manning approach, the conclusions should also acknowledge the method's limitations. For example, potential sources of uncertainty—such as assumptions of floodplain homogeneity, sensitivity to field-estimated parameters, and the challenges of validating results in data-sparse settings—deserve mention. Including both the advantages and constraints would provide a more balanced and complete perspective and help guide future applications and developments of the method.

*Our goal was to maintain a short Conclusions section in this Technical Note. However, the reviewer makes some good points here and we will consider expanding it and clarifying some of the points here and above.*

*7. Recommendations*
Reject

*Thanks for the recommendation. We hope that the above comments and revised manuscript can change this.*

*REFERENCES*

*Carling, P. A., Cao, Z., Holland, M. J., Ervine, D. A., & Babaeyan‑Koopaei, K. (2002). Turbulent flow across a natural compound channel. Water resources research, 38(12), 6-1.*

*Dietrich, W. E., Smith, J. D., & Dunne, T. (1979). Flow and sediment transport in a sand bedded meander. The Journal of Geology, 87(3), 305-315.*

*Dunne, K. B., & Jerolmack, D. J. (2020). What sets river width?. Science advances, 6(41), eabc1505.*

*Liu, J., Xiao, Y., Yuan, S., Zhang, T., Lin, Q., Yuan, K., ... & Gualtieri, C. (2025). Floodplain hydrodynamics and connectivity in a natural compound channel during unsteady flow events. Journal of Hydrology, 659, 133305.*

*Myers, W. R. C., & Brennan, E. K. (1990). Flow resistance in compound channels. Journal of Hydraulic Research, 28(2), 141-155.*

*Parker, G. (1978). Self-formed straight rivers with equilibrium banks and mobile bed. Part 2. The gravel river. Journal of Fluid mechanics, 89(1), 127-146.*

*Trampush, S. M., Huzurbazar, S., & McElroy, B. (2014). Empirical assessment of theory for bankfull characteristics of alluvial channels. Water Resources Research, 50(12), 9211-9220.*

---

## Author Comment (AC3)

I was Anonymous Referee #1 for the previous submission of this paper (https://egusphere.copernicus.org/preprints/2024/egusphere-2023-3118/). The paper has been reworked so that the proposed DoubleManning approach (unchanged) is better positioned among other rating curve methods reported in the literature. Another reason for commending the authors is that the paper writing and presentation are excellent. Data and software codes are made available.

*We send our thanks for your re-review and for the positive notes here.*

The rationale of the proposed method is meaningful. However I'm still sceptical about its novelty and practicality: the Divided Channel approach is classic (cf. the plentiful literature on compound channel flows), as well as segmented rating curves (eg. Rantz et al. 1982 for USGS procedures). As now stated in the article, Bayesian methods allow for the prior specification of the physical parameters of channel controls, similar to what is done here (arguably in a more natural and simple way), to constrain and compare the results. And they account for data uncertainty and provide result uncertainty, a big difference not really stressed in this paper…

*We also fundamentally agree with you here. Furthermore, we hope that our responses to comments by Anonymous Reviewer 1 provide additional background to some of the benefit in including geomorphological constraints – which we argue here provide real help in understanding and interpreting the results. We see this as a proposal of a formulation. If the formulation proves valuable, then the next step can be to incorporate it into a more powerful inversion method.*

In brief, in spite of the lack of novelty and practical usefulness, the offered method and tool are possibly worth a technical note, at least for the discussion of interesting ideas. I provide the following comments that may call for some improvements of the text and arguments.

*Thank you for the support, and for "in spite of the lack of novelty and practical usefulness", which made me smile. I argue that including some standard fluvial geomorphic principles and measurements is (unfortunately) novel when considering rating-curve development, and the double-Manning approach translates this conceptual model into a field-ready, if somewhat straightforward (though this is not always bad), numerical solution.*

The addition of two Manning equations does not account for head losses due to friction between main channel flow and floodplain flow. This effect is included in most 1D hydraulic models for instance. At least a discussion of this approximation and its possible consequences would be interesting.

*Interactions between flows in the main channel and those on the floodplain are good point to consider when computing flow velocities along different points in the cross-channel orientation. Momentum will, as I think you imply, mix between the slow floodplain flow and the fast channel flow, speeding the former and slowing the latter. Because our goal is to solve for overall discharge, these exchanges balance, though the faster near-bank floodplain flow can therefore experience more frictional resistance. We will make note of this assumption.*

It is unclear how (obvious) correlations in parameter estimates are handled: depending on the studied case, some parameters are fixed and others are let free to be calibrated. This appears to require some significant expertise from the user.

*Yes: The user is expected to be familiar with open-channel hydraulics and fluvial geomorphology, and to ideally have obtained data to help inform the modeling (e.g., slope, channel width, bankfull depth, bed-material grain size). The data obtained can inform which parameters to constrain and which to vary.*

The 3 examples are useful but they also show that the use of the method is not straightforward, as it requires adapting the modelling strategy to the amount of information available in the data and in the prior knowledge on channel and floodplain. Summarizing a general workflow applicable to any situation would be useful, in the end of the paper.

*Summarizing a workflow does seem like a good idea. If it does not make this Technical Note too long, we refer users to the guide that we have written at https://github.com/MNiMORPH/doublemanning.*

A decisive advantage of modelling the channel and floodplain as separate terms in the rating curve equation (i.e. the divided channel approach, as opposed to modelling the whole river section as a single term) is that the calibration of the main channel parameters still holds for overbank flows, which reduces the discharge uncertainty even when few or no discharge measurement exist for overbank flows. I have not seen such an argument in the paper.

*This is a good point, and we did not think of it as something worthy to mention. We will include it. Thank you for the help in making the paper stronger.*